# Switching of RNA splicing regulators in immature neuroblasts during adult neurogenesis

Corentin Bernou[1,2], Marc-André Mouthon[1,2], Mathieu Daynac[1,2], Thierry Kortulewski[1,2], Benjamin Demaille[1,2], Vilma Barroca[1,2], Sebastien Couillard-Despres[3,4,5], Nathalie Dechamps[1,2], Véronique Ménard[1,2], Léa Bellenger[6], Christophe Antoniewski[7], Alexandra Déborah Chicheportiche[1,2]*, François Dominique Boussin[1,2]*

[1]Université Paris Cité, Inserm, CEA, Stabilité Génétique Cellules Souches et Radiations, LRP/iRCM, Fontenay-aux-Roses, France; [2]Université Paris-Saclay, Inserm, CEA, Stabilité Génétique Cellules Souches et Radiations, LRP/iRCM, Fontenay-aux-Roses, France; [3]Spinal Cord Injury and Tissue Regeneration Center Salzburg (SCI-TReCS), Paracelsus Medical University, Salzburg, Austria; [4]Institute of Experimental Neuroregeneration, Paracelsus Medical University, Salzburg, Austria; [5]Austrian Cluster for Tissue Regeneration, Vienna, Austria; [6]Inserm, ARTbio Bioinformatics Analysis Facility, Sorbonne Université, CNRS, Institut de Biologie Paris Seine, Paris, France; [7]ARTbio Bioinformatics Analysis Facility, Sorbonne Université, CNRS, Institut de Biologie Paris Seine, Paris, France

*For correspondence:
alexandra.chicheportiche@cea.fr (ADC);
francois.boussin@cea.fr (FDB)

Competing interest: The authors declare that no competing interests exist.

## eLife assessment

This **useful** manuscript presents an intriguing potential refinement of models for adult SVZ neurogenesis, and highlights the role of RNA splicing at specific stages in the lineage. Unfortunately, the evidence does not fully support the claims, leaving it currently **incomplete**. The proposed role of RNA splicing in neuronal differentiation, though interesting, remains unexplored and would benefit significantly from targeted gene manipulation studies.

**Abstract** The lateral wall of the mouse subventricular zone harbors neural stem cells (NSC, B cells) which generate proliferating transient-amplifying progenitors (TAP, C cells) that ultimately give rise to neuroblasts (NB, A cells). Molecular profiling at the single-cell level struggles to distinguish these different cell types. Here, we combined transcriptome analyses of FACS-sorted cells and single-cell RNAseq to demonstrate the existence of an abundant, clonogenic and multipotent population of immature neuroblasts (iNB cells) at the transition between TAP and migrating NB (mNB). iNB are reversibly engaged in neuronal differentiation. Indeed, they keep molecular features of both undifferentiated progenitors, plasticity and unexpected regenerative properties. Strikingly, they undergo important progressive molecular switches, including changes in the expression of splicing regulators leading to their differentiation in mNB subdividing them into two subtypes, iNB1 and iNB2. Due to their plastic properties, iNB could represent a new target for regenerative therapy of brain damage.

## Introduction

New neurons are constantly produced from neural stem cells (NSC) throughout life in two restricted neurogenic areas of the adult mammalian brain: the sub-granular zone (SGZ) of the hippocampus and the subventricular zone (SVZ; *Obernier and Alvarez-Buylla, 2019*). The lateral wall (LW) of the SVZ, adjacent to the striatum, is the largest germinal zone of the adult mouse brain, in which NSC (type B1 cells) are produced during the embryonic development and are mostly in a quiescent state thereafter (*Codega et al., 2014*; *Tong et al., 2015*). Upon activation, NSC divide and give rise to rapidly dividing transit-amplifying cells (TAP, type C cells), which ultimately produce neuroblasts (NB, type A cells). These neuroblasts migrate out of the SVZ anteriorly along blood vessels and the rostral migratory stream (RMS) to reach the olfactory bulb (OB) where they integrate and differentiate into various subtypes of interneurons required to fine-tune odor discrimination (*Obernier and Alvarez-Buylla, 2019*). The diversity of the OB interneurons depends on the intrinsic regional identity of NSC in the SVZ along the anterior-posterior and medial-lateral axes (*Obernier and Alvarez-Buylla, 2019*; *Merkle et al., 2014*).

Besides their neurogenic fate, NSC and neural progenitors present in the SVZ are also proficient to produce astrocytes and oligodendrocytes that migrate toward the corpus callosum, the RMS and the striatum in the normal (*Gonzalez-Perez and Alvarez-Buylla, 2011*; *Hack et al., 2005*; *Sohn et al., 2015*) and injured adult brain (*Holmin et al., 1997*; *Nait-Oumesmar et al., 1999*).

Transcriptomic analyses using different combinations of cell surface markers and transgenic mice models to isolate SVZ cells have provided important initial insights into stem cell quiescence and have highlighted consecutive transitions during neuronal lineage progression (*Codega et al., 2014*; *Belenguer et al., 2021*; *Fischer et al., 2011*; *Mich et al., 2014*; *Pastrana et al., 2009*). Since then, the single-cell RNA-sequencing has revolutionized the field and has made it possible to precisely elucidate the transcriptome of SVZ cells present in the LW (*Cebrian-Silla et al., 2021*; *Dulken et al., 2017*; *Xie et al., 2020*; *Zywitza et al., 2018*) and in the septal wall which also harbours NSC niches (*Mizrak et al., 2019*).

We have previously developed a FACS-based method, using a combination of three surface markers (LeX, EGF receptor, CD24), allowing to sort five distinct SVZ cell populations: quiescent NSC (LeX-EGFR-CD24-, hereafter referred to as sorted-qNSC, s-qNSC), activated NSC (LeX +EGFR + CD24-, referred to as sorted, s-aNSC), transit amplifying cells (LeX⁻EGFR⁺CD24⁻, referred to as sorted, s-TAP), immature (LeX⁻EGFR⁺CD24$^{low}$, referred to as sorted, s-iNB) and migrating neuroblasts (LeX⁻EGFR⁻CD24$^{low}$, referred to as sorted, s-mNB) from the adult mouse brain (*Daynac et al., 2015*; *Daynac et al., 2013*). Importantly, whereas s-iNB presented neuroblast markers, for example CD24 and doublecortin (DCX), s-iNB gathered the most abundant population among SVZ cycling progenitors (representing approximately 4.3% of total SVZ cells *vs* 1% s-aNSC and 2.5% s-TAP; *Daynac et al., 2013*).

We have previously shown that following a genotoxic stress (exposure to ionizing radiation) inducing the depletion of cycling progenitors, s-qNSC exit quiescence restoring successively s-aNSC, s-TAP, s-iNB and finally s-mNB (*Daynac et al., 2013*).

Transcriptomic studies revealed that age-related SVZ neurogenesis decline is associated with TGFß signaling and specific cell cycle regulation changes in s-aNSC. A progressive lengthening of aNSC G1 phase with age leads to a decrease in the production of s-TAP, s-iNB, and s-mNB (*Daynac et al., 2013*; *Daynac et al., 2016*; *Pineda et al., 2013*). However, despite the decrease of neurogenesis with aging the s-iNB remained more abundant than s-aNSC and s-TAP in the aged brain (*Daynac et al., 2016*).

Studies of SVZ neural progenitors have been essentially focused on the characterization of the molecular mechanisms regulating quiescence, activation and the regenerative potential of aNSC. Here, we combined transcriptome analyses using DNA microarrays on FAC-sorted cells and single-cell RNAseq to demonstrate that iNB form a molecularly distinct class of SVZ cells, exhibiting molecular features of both neural progenitors and neuroblasts while keeping unexpected regenerative capacity and plasticity. We identified important and sequential molecular switches occurring in these cells before their final differentiation in migrating neuroblasts. Altogether, these data led us to propose a revision of the current model of adult neurogenesis.

# Results

## Regenerative potential and multipotency of sorted iNB in vitro and in vivo

We have previously shown that s-iNB, which expressed the neuroblast markers CD24 and DCX (*Daynac et al., 2013*) and markers of neural progenitors such as Mash1 (*Daynac et al., 2013*), Dlx2 and SOX2 (*Figure 1—figure supplement 1*) had a clonogenic potential and were able to proliferate in vitro for weeks in normoxic conditions (20%O2) unlike s-mNB (*Daynac et al., 2013*). However, s-iNB exhibited reduced clonogenic potential compared to that of s-aNSC and s-TAP (*Figure 1A*). It has been previously reported that hypoxic conditions promoted SVZ NSC proliferation and neurogenesis (*Li et al., 2021*; *Ross et al., 2012*). Hypoxic conditions (4%O2) had no effects on s-mNB, but increased the proliferation capacities of s-aNSC, s-TAP and s-iNB, which reached similar growth rates, making them undistinguishable based on this endpoint (*Figure 1A*). Hypoxic conditions also increased the clonogenic potentials of both s-TAP and s-iNB but not that of s-aNSC (*Figure 1B*). In these conditions, the clonogenic potential of s-TAP reached similar values as s-aNSC, but that of s-iNB was still lower (*Figure 1B*).

Quite surprisingly, freshly isolated s-iNB had the capacity to generate the three neural lineages, neurons, astrocytes and oligodendrocytes, when plated for 5–7 days in the appropriate differentiation media, similarly as s-aNSC and s-TAP (*Figure 1C*).

Thus, to assess the regenerative potential of these immature neuroblasts in vivo, eGFP$^+$ iNB were sorted from β−actin:eGFP mice, transplanted into the left striatum of recipient C57Bl/6 mice and compared to eGFP$^+$s-mNB and eGFP$^+$s-NSC/TAP. Five weeks after transplantation, eGFP$^+$ cells were detected in different regions of interest (*Table 1*) in the brains of mice transplanted with eGFP$^+$s-NSC/TAP or eGFP$^+$s-iNB, whereas no eGFP +were observed after transplantation of eGFP$^+$s-mNB even at the injection site (IS) (*Table 1*). Excepted in one out of three, animals transplanted with eGFP$^+$s-iNB presented high levels of eGFP$^+$ cells in the OB with levels equivalent or superior to animals transplanted with eGFP$^+$s-NSC/TAP or eGFP$^+$s-iNB (*Figure 1D*). EGFP$^+$ cells were detected in the RMS (*Figure 1—figure supplement 2*) as well as in the granular cell layer (GCL) and the external plexiform layer (EPL) of the OB where they differentiated into $\mathrm{N}$euN$^+$ neurons in 2/3 animals transplanted with eGFP$^+$s-iNB (*Figure 1Da*) and in 3/3 animals transplanted with EGFP$^+$s-NSC/TAP (*Table 1*).

Importantly, eGFP$^+$ cells were present in the SVZ of all the animals transplanted with eGFP$^+$s-iNB and eGFP$^+$s-NSC/TAP (*Figure 1Db*, *Figure 1Dc*), some of them expressing GFAP indicating the generation of astrocytes, and therefore possibly NSC (*Figure 1Db3*, *Figure 1Dc2*). Plasticity of s-iNB was also shown at the IS by the detection of eGFP$^+$ cells expressing markers of astrocytes (GFAP) and oligodendrocytes (CNPase; *Figure 1—figure supplement 2B and C*).

Overall, these results demonstrate the plasticity of s-iNB in vitro, making them very similar to s-NSC/TAP, questioning the exact nature of this abundant SVZ cell populations that express neuroblast markers.

## s-iNB are a molecularly distinct class of proliferating SVZ progenitors

We characterized the transcriptomic profiles of each SVZ populations sorted from 2-month-old C57Bl/6 mice using Clariom D mouse microarrays that provides intricate transcriptome-wide gene- and exon-level expression profiles (*Figure 2A*).

Pseudotime analysis (TSCAN; *Ji and Ji, 2016*) based on 3679 highly dispersed genes between the five sorted cell populations confirmed the lineage progression from s-qNSC (cluster1), then s-aNSC and s-TAP within a single cluster (cluster2), followed by s-iNB (cluster3) and finally s-mNB (cluster4; *Figure 2B*). Another pseudotime analysis excluding s-qNSC, based on 1481 highly dispersed genes, delineated two distinct clusters for s-aNSC and s-TAP (*Figure 2C*), highlighting their molecular differences.

Venn diagram comparison of data sets revealed 2950, 315, 129, 395, and 1693 genes unique to s-qNSC, s-aNSC, s-TAP, s-iNB, and s-mNB respectively (*Figure 2—figure supplement 1—source data 1*). G:Profiler functional profiling of these transcriptome signatures with KEGG and reactome databases uncovered significant stage-dependent changes (*Figure 2—figure supplement 1—source data 1*). S-aNSC overexpressed genes involved in metabolism of glyoxylate and dicarboxylate metabolism. S-qNSC rather overexpressed genes related to metabolism of lipids and Rho GTPase cycle. S-TAP significantly displayed genes involved in Mitotic, G1 phase and G1/S transition, Activation of

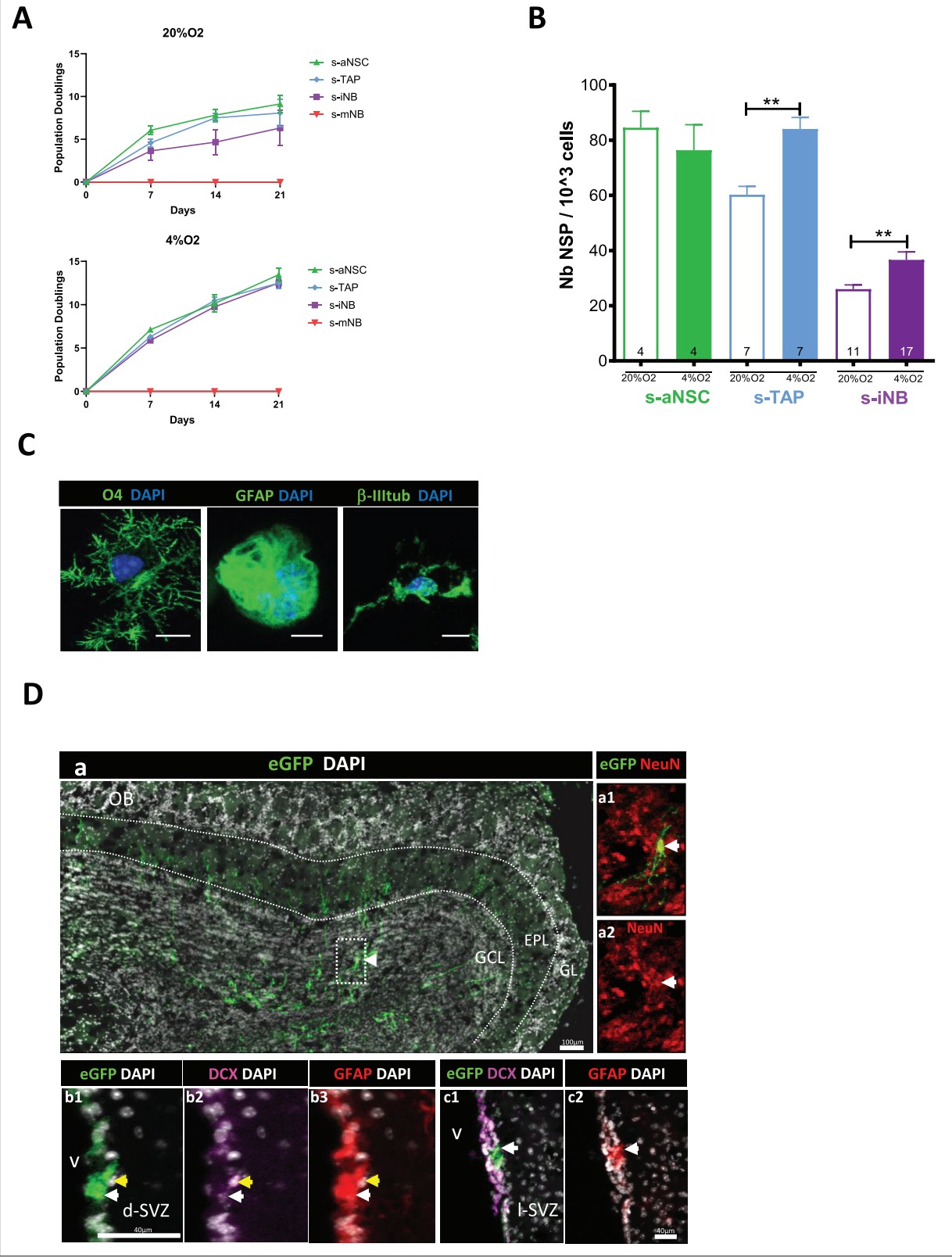

**Figure 1.** Plasticity of iNB in vitro and in vivo: s-aNSC, s-TAP, s-iNB, and s-mNB were sorted from the SVZ of 2-month-old C57Bl/6 mice. (**A**) Population doublings (PD). Data were obtained from three independent experiments (n=8). (**B**) Clonogenicity assay. Results were obtained from two independent experiments. The number inside the bars indicated the number of microplate wells analyszed (mean ± SEM). (**C**) Differentiation assay. Representative images of immunofluorescence of freshly sorted iNB cells cultured in oligodendrocyte, astrocyte, or neuronal differentiation medium, and stained for

*Figure 1 continued on next page*

*Figure 1 continued*

NG2 and CNPase, GFAP and CD133 or βIII-Tubulin and doublecortin (DCX) expressions, respectively. *p<0.05, **p<0.01, ***p<0.001 (Mann-Whitney U-test). Scale bar: 20 µm. (**D**) EGFP-positive s-iNB were isolated from β–actin:eGFP mice and transplanted unilaterally at three injection points at proximity of the dSVZ/RMS of recipient C57Bl/6 J mice. Transplanted brains were analyszed 5 weeks later by immunostaining. (**a**) Detection of eGFP + cells in the granule cell layer and the external cell layer of the olfactory bulb of a mouse transplanted. (a1–a2) a2 high magnification of the inset (dotted line in a) showing eGFP + NeuN + cells (white arrows). (**b–c**) Detection of eGFP⁺ GFAP⁺ DCX⁺ cells (yellow arrows) and eGFP +GFAP + (white arrows) in the dorsal (**b2**) and lateral (**c2**) SVZ. GCL:granule cell layer, EPL: external plexiform layer, GL: glomerular layer, lSVZ: lateral SVZ, dSVZ: dorsal SVZ, Scale bars = 40 µm or 100 µm.

The online version of this article includes the following figure supplement(s) for figure 1:

**Figure supplement 1.** Expression of neural progenitor markers by SVZ cell-sorted populations.

**Figure supplement 2.** eGFP⁺ s-iNB cells in the rostral migratory stream and regenerative potential of eGFP +s iNB in vivo.

the pre-replicative complex. S-mNB specifically overexpressed genes involved in axon guidance and nervous system development. Conversely, s-iNB signatures contained many genes linked to cell cycle, prometaphase, M phase, and cell cycle checkpoints, clearly distinguishing them from the other cell populations including s-mNB.

Alternative splicing (AS) plays various functions in brain development including neural cell differentiation and neuronal migration (*Fisher and Feng, 2022*; *Grabowski, 2011*; *Jacko et al., 2018*; *Weyn-Vanhentenryck et al., 2018*; *Yano et al., 2010*). Using the splicing index (SI) corresponding to a value assigned to each exon by estimating its abundance with respect to adjacent exons using TAC software (Thermo Fisher Scientific), we identified differentially spliced genes (DSG, adjusted p-value ≤0.05) unique to s-qNSC, s-aNSC, s-TAP, s-iNB. and s-mNB (*Figure 2—figure supplement 2—source data 1*). GO annotations of DSG clearly revealed that spliced genes in SVZ cells are mainly involved in neuron development and neurogenesis. Interestingly, this also showed that qNSC logically differed from the other cell types by splicing concerning genes involved in mitosis and cell cycle, consistently with their quiescent state. More importantly, GO annotations of DSG confirmed that s-TAP and s-iNB have distinct features.

**Table 1.** Transplantations of eGFP⁺s-iNB, eGFP⁻s-iNB and eGFP⁺s-mNB freshly isolated from β-actin:eGFP mice model.
Table recapitulating the immunohistological analyses of the numbers of eGFP⁺ cells in different regions in recipient C57Bl6/J mice brains, 5 weeks after transplantation.

| | | IS | | SVZ | | CC | | OB | | Total eGFP + cells |
|---|---|---|---|---|---|---|---|---|---|---|
| | | Nb eGFP + cells | % | Nb eGFP + cells | % | Nb eGFP + cells | % | Nb eGFP + cells | % | |
| s-aNSC/s-TAP | Animal 1 | 271 | 62.6 | 17 | 3.9 | 55 | 12.7 | 90 | 20.8 | 433 |
| 1.3–1.5x10⁴ | Animal 2 | 300 | 31.9 | 79 | 8.4 | 47 | 5.0 | 515 | 54.7 | 941 |
| eGFP + engrafted cells | Animal 3 | 95 | 37.1 | 73 | 28.5 | 22 | 8.6 | 66 | 25.8 | 256 |
| s-iNB | Animal 1 | 1025 | 65.2 | 16 | 1.0 | 49 | 3.1 | 483 | 30.7 | 1573 |
| 2x10⁴ | Animal 2 | 196 | 15.5 | 41 | 3.2 | 41 | 3.2 | 989 | 78.1 | 1267 |
| eGFP + engrafted cells | Animal 3 | 240 | 84.8 | 43 | 15.2 | 0 | 0 | 0 | 0 | 283 |
| s-aNSC/s-TAP | Animal 1 | 0 | 0 | 0 | 0 | 0 | 0 | 0 | 0 | 0 |
| 1.3–1.5x10⁴ | Animal 2 | 0 | 0 | 0 | 0 | 0 | 0 | 0 | 0 | 0 |
| eGFP + engrafted cells | Animal 3 | 0 | 0 | 0 | 0 | 0 | 0 | 0 | 0 | 0 |

IS = Injection Site; CC = Corpus callosum; SVZ = Subventricular zone; OB = Olfactive bulb.

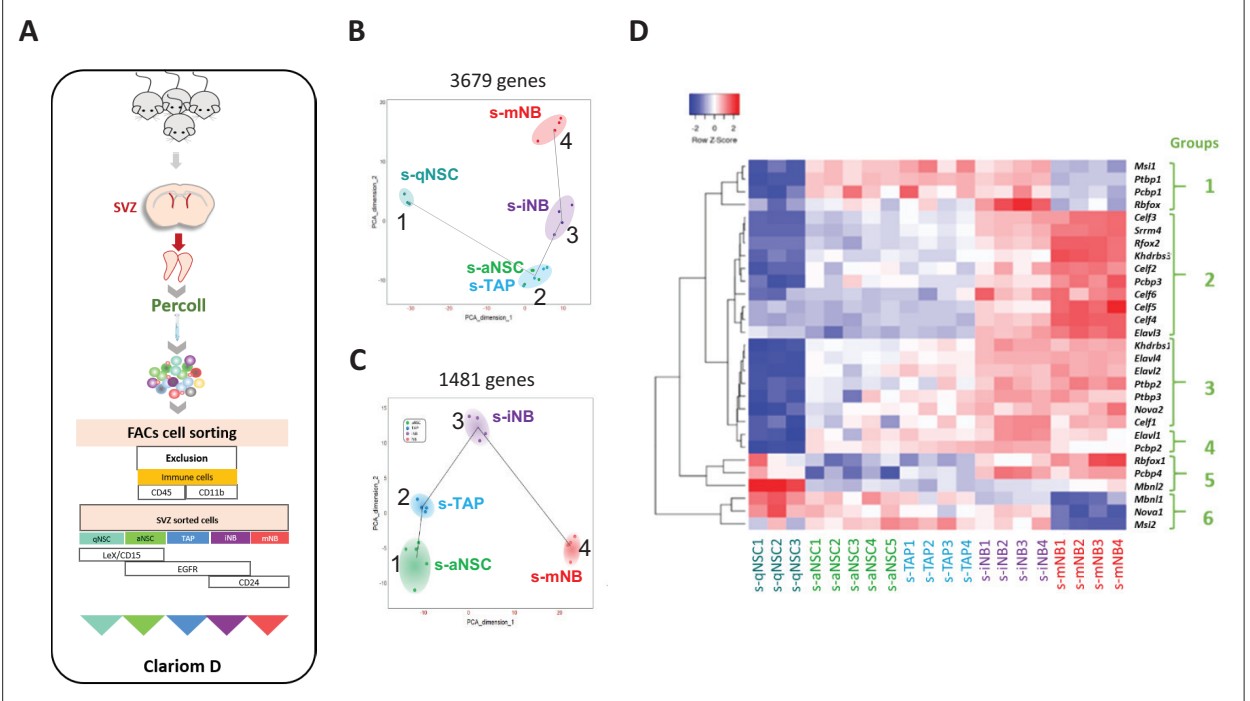

**Figure 2.** Transcriptomic analysis of SVZ neural progenitor cells. (**A**) Schematic illustration of our strategy of flow cytometry cell sorting of the five major neurogenic populations (s-qNSC, s-aNSC, s-TAP, s-iNB, s-mNB) of SVZ microdissected from adult mouse brains. (**B**) Pseudotime analysis showing the lineage progression of sorted cell populations based on the highly dispersed genes between s-qNSC, s-aNSC, s-TAP, s-iNB, and s-mNB. (**C**) Pseudotime analysis after exclusion of s-qNSC based to further discriminate s-aNSC and s-TAP. (**D**) Heatmap representation of RSR genes differentially expressed among s-aNSC, s-TAP, s-iNB, s-mNB (FDR p-value <0.05).

The online version of this article includes the following source data and figure supplement(s) for figure 2:

**Figure supplement 1.** Functional enrichment analyses of the individual transcriptome signature of neurogenic populations.

**Figure supplement 1—source data 1.** Transcriptomic signatures of s-qNSC, s-aNSC, s-TAP, s-iNB and s-mNB (Clariom assay) and corresponding g:Profiler Gene Ontology analyses.

**Figure supplement 2.** Enrichment of GO terms corresponding to mRNA splicing isoforms in the different types of sorted SVZ cells.

**Figure supplement 2—source data 1.** Differentially spliced genes (DSG) unique to s-qNSC, s-aNSC, s-TAP, s-iNB, and s-mNB and corresponding Gene Enrichment Analyses (GSEA).

**Figure supplement 2—source data 2.** Differentially spliced genes in each transition along SVZ neurogenesis and corresponding g.Profiler Gene Ontology analyses.

**Figure supplement 2—source data 3.** Differentially expressed genes (DEG) among RNA splicing regulators (RSR) genes in each transition along SVZ neurogenesis.

**Figure supplement 3.** Correspondence between the molecular signatures of sorted neural progenitor populations in our study and available sc-RNA-seq datasets. Violin diagrams representing the expression of selected clusters from published sc-RNAseq analysis in our Clariom microarray(14,17,32). S-qNSC perfectly matched the expression of Glial markers (Class III)(32), qNSC(17) and Quiescent B cells (clusters 5 and 14)(14). In contrast, s-aNSC poorly matched with Lipid biosynthesis (Class III) and Cell cycle (Class IV)(32), aNSC(17) and Activated B cells (Cluster 13)(14). Similarly, s-TAP poorly matched with TAP(17) and C cells (Cluster 12)(14). S-iNB perfectly fit with Mitosis (Class VI), mitotic TAP (mTAP) (Zywitza et al., 2018) and Mitosis (Cluster 8, 10,16 and 17)(14). s-mNB overlapped genes related to Neuron differentiation (Class VII), Late NBs (17) and A cells (clusters 0, 1 and 4).

We further analyszed particularly DSG at the successive cell transitions along neurogenesis. Results show a very large number of DSG between s-qNSC *vs* s-aNSC (9484) then no DSG at the s-aNSC and s-TAP transition, only 169 between the s-TAP and s-iNB transition, and finally 3759 between s-iNB and s-mNB (*Figure 2—figure supplement 2—source data 2*). Functional profiling analysis with g:Profiler linked the DSG between s-TAP and s-iNB to the nervous system development and those between s-iNB *vs* s-mNB to protein binding or cell junction.

We then focused on the differentially expressed genes (adjusted p-value ≤0.05) belonging to the nine families of RNA splicing regulators (RSR) that play essential roles in the neurogenic pathway (*Ptbp, Nova, Rbfox, Elavl, Celf, Dbhs, Msi, Pcbp, and Mbnl*; *Fisher and Feng, 2022*). Strikingly, a clear

down-regulation of most RSR gene expression distinguishes s-qNSCs from the other populations (*Figure 2—figure supplement 2—source data 3*). Interestingly, another shift discriminates s-aNSC and s-TAP from s-mNB (*Figure 2—figure supplement 2—source data 3*). This led us to identify several groups of RSR genes in function of their variation of expression along with neurogenesis: the groups 1 and 6 were overexpressed in s-aNSC and s-TAP and downregulated in s-mNB; the group 2, 3, and 5 were up-regulated in mNB as compared to s-aNSC and s-TAP. All these groups were up-regulated in s-iNB, suggesting that the switch in RSR expression between cycling progenitors and s-mNB occurred in these cells (*Figure 2D*). Interestingly, *Ptbp1*, encoding a major splicing repressor (*Zhu et al., 2020*), was downregulated in s-mNB as compared to s-iNB, consistently with the large increase in DSG at the transition between s-iNB and s-mNB.

Altogether, these data confirm at the molecular level that s-iNB comprises cycling progenitors at the transition between s-TAP and s-mNB that undergo important molecular switches, which include major changes in the expression of some RSR genes leading to a major increase in the number of DSG occurring at the final maturation of s-mNB.

## s-iNB correspond to specific SVZ cell clusters identified by single-cell RNA sequencing (scRNAseq)

We compared our microarray datasets to published scRNAseq datasets from adult mouse SVZ (*Cebrian-Silla et al., 2021*; *Zywitza et al., 2018*; *Llorens-Bobadilla et al., 2015*). Violin plots show that s-qNSC and s-mNB perfectly matched with the corresponding populations identified in these studies, which was less obvious for s-aNSC and s-TAP (*Figure 2—figure supplement 3*). s-iNB matched with the clusters termed 'Mitosis' (Class VI) in the study of *Llorens-Bobadilla et al., 2015* and 'Mitotic TAP' in that of *Zywitza et al., 2018*, and with clusters 8, 10, 17, 16 linked to 'Mitosis', but also with clusters 6 and 15 described as DCX⁺Ki67⁺ neuroblasts in the study by *Cebrian-Silla et al., 2021*. These comparisons thus confirm that s-iNB comprise cycling SVZ cells exhibiting both TAP and NB features, which illustrates the difficulties to determine the exact nature of cell clusters identified by scRNAseq.

We have previously reported (*Daynac et al., 2013*) that after brain exposure to ionizing radiation, proliferating progenitors are highly radiation-sensitive and rapidly eliminated after irradiation. On the contrary qNSC that are radiation resistant repopulate the different SVZ neurogenic cell populations successively the following days after irradiation. We therefore used this model and our transcriptomic signatures of SVZ stem and progenitor cells to gain new insights on the nature and the hierarchical ordering of neural progenitor clusters identified by sc-RNAseq.

TenX Genomics droplet-based single-cell transcriptomic analyses were performed on SVZ cells from two mice, 5 days after irradiation (4 Gy, SVZ only) and from two non-irradiated control mice after removal of myelin and red blood cells (*Figure 3A*). Based on a filtering criterion of 1500 genes/cell to exclude low-quality sequenced cells, 17,343 cells were retained for analysis, divided into 11,529 non-irradiated cells and 5814 irradiated cells (*Figure 3—figure supplement 1*).

Non-irradiated and irradiated samples were combined and integrated using the SCTransform workflow. After integration, PCA was performed and 50 PCs were used for Uniform-manifold approximation (UMAP). Leiden graph-based clustering, with a resolution of 1.2, identified 33 clusters (*Figure 3—figure supplement 1A*). These clusters were first annotated based on the expression of known cell markers (*Figure 3—figure supplement 1B*). Nineteen clusters gathering more than half SVZ-residing cells corresponded to non-neurogenic cells (ordered by decreasing abundance): microglia (expressing *P2ry12 Gómez Morillas et al., 2021*), ependymal cells (expressing both *S100b* and *Slc1a3 Berger and Hediger, 1998*), endothelial cells (expressing *Flt1 Sawano et al., 2001*), oligodendrocyte progenitors (OPC, expressing *Pdgfra Zhu et al., 2014*), Myelin forming oligodendrocytes (MFOL, expressing *Mog Zywitza et al., 2018*) and pericytes (expressing both *Vtn* and *Pcam1* (*Sohn et al., 2015*; *Lertkiatmongkol et al., 2016*), and dopamine neurons (expressing *Slc17a6 Kouwenhoven et al., 2020*). The 14 other clusters corresponded to astrocytes and neurogenic cells (*Figure 3B–C–D*). The clusters 4 and 13 that expressed *S100b* were further identified as qNSC (cluster 4) and astrocytes (cluster 13) based on the transcriptomic signatures described by *Cebrian-Silla et al., 2021*; *Figure 3—figure supplement 2*; *Figure 2—figure supplement 3*). Cluster 17, was identified as aNSC based on *Gfap, Thbs4, S100a6, Egfr^low* and *Ascl1* expressions (*Cebrian-Silla et al., 2021*), and cluster 20 as TAP based on *Egfr* and *Ascl1* expressions (*Kim et al., 2011*). Clusters 10, 5, 15, 12, and 8 were defined as cycling progenitors based on the expression of

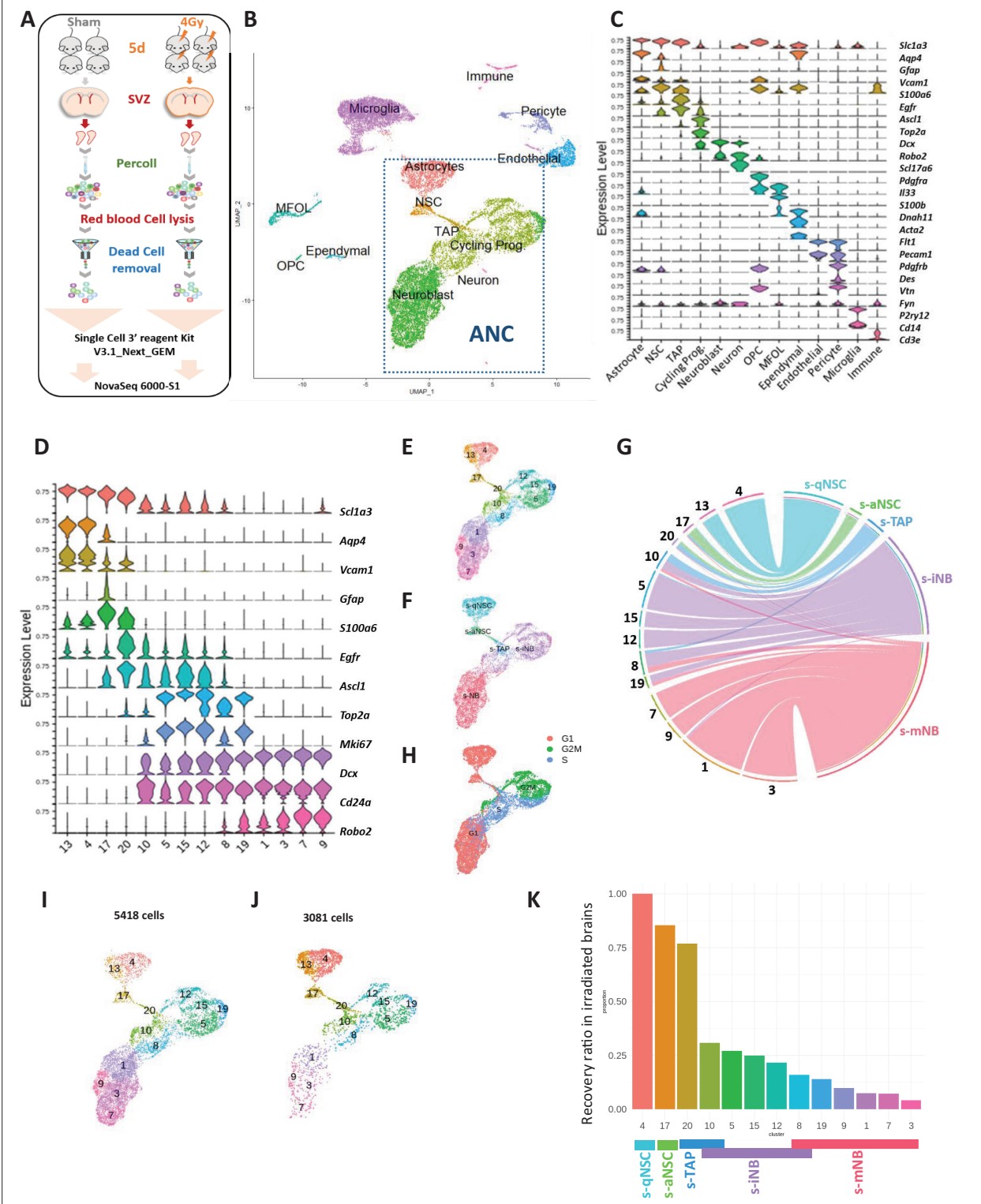

**Figure 3.** scRNA-seq analysis: whole SVZ. (**A**) Experimental design. (**B**) Uniform Manifold Approximation and Projection (UMAP) of the 17343 high-quality sequenced cells annotated with corresponding cell type, combining scRNA-seq datasets of non-irradiated mice and 4 Gy irradiated mice. Clustering analysis at resolution 1.2 segregated cells into 33 clusters that were matched with corresponding cell types determined with marker expression in (**C**). Among them, 14 clusters (dotted lines rectangle) corresponded to the clusters of astrocytes and neurogenic cells (ANC). (**C**) Violin diagrams illustrating the expression level of known cell markers in the 33 clusters according to the literature. (**D**) Violin diagrams representing the expression level of selected markers specific to neurogenic cells and astrocyte from ANC clusters at the resolution 1.2. (**E**) UMAP focusing on the ANC

*Figure 3 continued on next page*

*Figure 3 continued*

subset. (**F**) UMAP of the top UCell score for the top 100 highly expressed genes of each neurogenic cell sorted population in the ANC clusters (*Figure 2—figure supplement 1—source data 1*). (**G**) Chord diagram illustrating the correspondence of cell clusters with cell top score. Cluster 10 matched with both s-TAP and s-iNB and clusters 8 and 19 both to s-iNB and to s-mNB. (**H**) Feature plot of cell cycle scoring (**I, J**) UMAP representation of ANC clusters in the unirradiated (**I**) and irradiated (**J**) samples. (**K**) Barplot showing the recovery ratios of each ANC cluster in irradiated brains calculated as the number of cells per clusters from irradiated brains /control brains normalized to the respective numbers of s-qNSC (cluster 4). Taking into account UCell score results, s-iNB clusters overlapped the cluster 10 and 8 containing s-TAP and s-mNB, respectively.

The online version of this article includes the following figure supplement(s) for figure 3:

**Figure supplement 1.** Quality controls of scRNA-seq on whole SVZ tissues.

**Figure supplement 2.** Annotation of sc-RNAseq clusters and identification of astrocytes and qNSCs.

---

proliferative markers such as *Top2a*, *Mki67*, *Ascl1*. Clusters 1, 3, 7, and 9 were identified as mNB due to the loss of *Mki67*, *Top2 a* and *Ascl1* expressions and the expression of *Robo2* and *Dcx*. Cluster 19 did not expressed *Ascl1* but *Top2a* and *Mki67* as well as with *Robo2* and *Dcx,* was therefore positioned at the transition between iNB and mNB.

We then applied the UCell workflow to further characterize the neurogenic clusters with the transcriptomic signatures of each of our five sorted SVZ cell populations (*Figure 2—figure supplement 3*), calculating individual cell scores for each signature based on the Mann–Whitney U statistical analysis, and attributing cells to populations based on the highest score amongst signatures (*Andreatta and Carmona, 2021*). This score allowed us to unequivocally attribute some clusters to the sorted cell populations (*Figure 3E, F and G*): cluster 4 and 13 matched with s-qNSC, cluster 17 with s-aNSC, cluster 20 with s-TAP, clusters 5, 15 and 12 with s-iNB and finally clusters 7, 9, 1 and 3 to s-mNB. Reflecting the continuous nature of the SVZ neurogenic process, the cluster 10 matched both with s-TAP and s-iNB and clusters 8 and 19 both to s-iNB and to s-mNB. Cell-Cycle Scoring (*Zhu et al., 2014*) indicated that the clusters related to s-iNB were enriched in cells in S or G/2 M phase (*Figure 3H*).

As the consequence of radiation-induced cell death, the relative abundances of each cluster were different in controls (*Figure 3I*) and irradiated (*Figure 3J*) brains and dependent on the progressive repopulation of the different types of progenitors. Normalized to the numbers of s-qNSC (cluster 4), which are supposed not to be affected by irradiation (*Daynac et al., 2013*), the ratios of the number of cells per clusters in irradiated brains over that in unirradiated controls gave an index of the progressive reconstitution of each cluster (*Figure 3K*). As expected, clusters 17 and 20, corresponding to s-aNSC and s-TAP respectively, were almost entirely regenerated. The decreasing values of the ratios evidence a temporal ordering of the regeneration of each cluster corresponding to s-iNB along with neurogenesis as follows: 10, 5, 15, 12, and 8, the latter being the one at the iNB vs mNB transition as anticipated above.

We next addressed the expression of RSR genes in each cluster (*Figure 4*). Not all genes could have been investigated since some of them were below the detection threshold in all the cell populations studied. The expression of RSR genes (*Kdrbs3, Kdrbs1 Elavl3 and Elavl2, Elavl4, and Celf1*) that were shown to increase along with neurogenesis in microarrays, also increased in cluster 10, remained stable in the other clusters related to s-iNB and reached a maximum value in the s-mNB clusters (*Figure 4A*). Conversely, the expressions of *Mbnl1* and *Msi2* that were shown to decrease along with neurogenesis in microarrays (*Figure 4B*) peaked in the clusters related to s-aNSC (*Zywitza et al., 2018*) and s-TAP (*Daynac et al., 2013*), decreased in the cluster 10, remained low in the clusters related to s-iNB (*Gonzalez-Perez and Alvarez-Buylla, 2011*; *Dulken et al., 2017*; *Mich et al., 2014*; *Holmin et al., 1997*) and were not detected in the clusters related to s-mNB (*Daynac et al., 2015*; *Obernier and Alvarez-Buylla, 2019*; *Tong et al., 2015*; *Sohn et al., 2015*; *Nait-Oumesmar et al., 1999*). Altogether, these data corroborate our microarray data and show that a major switch in the expression of RSR genes occurs in the clusters related to s-iNB. Interestingly, the splicing repressor *Ptbp1* was similarly expressed in s-aNSC, s-TAP and s-iNB-related clusters but was downregulated progressively in the successive s-mNB clusters, suggesting that changes in *Ptbp1* expression constitute an important molecular step for the final maturation of mNB, which involves a major increase of DSG.

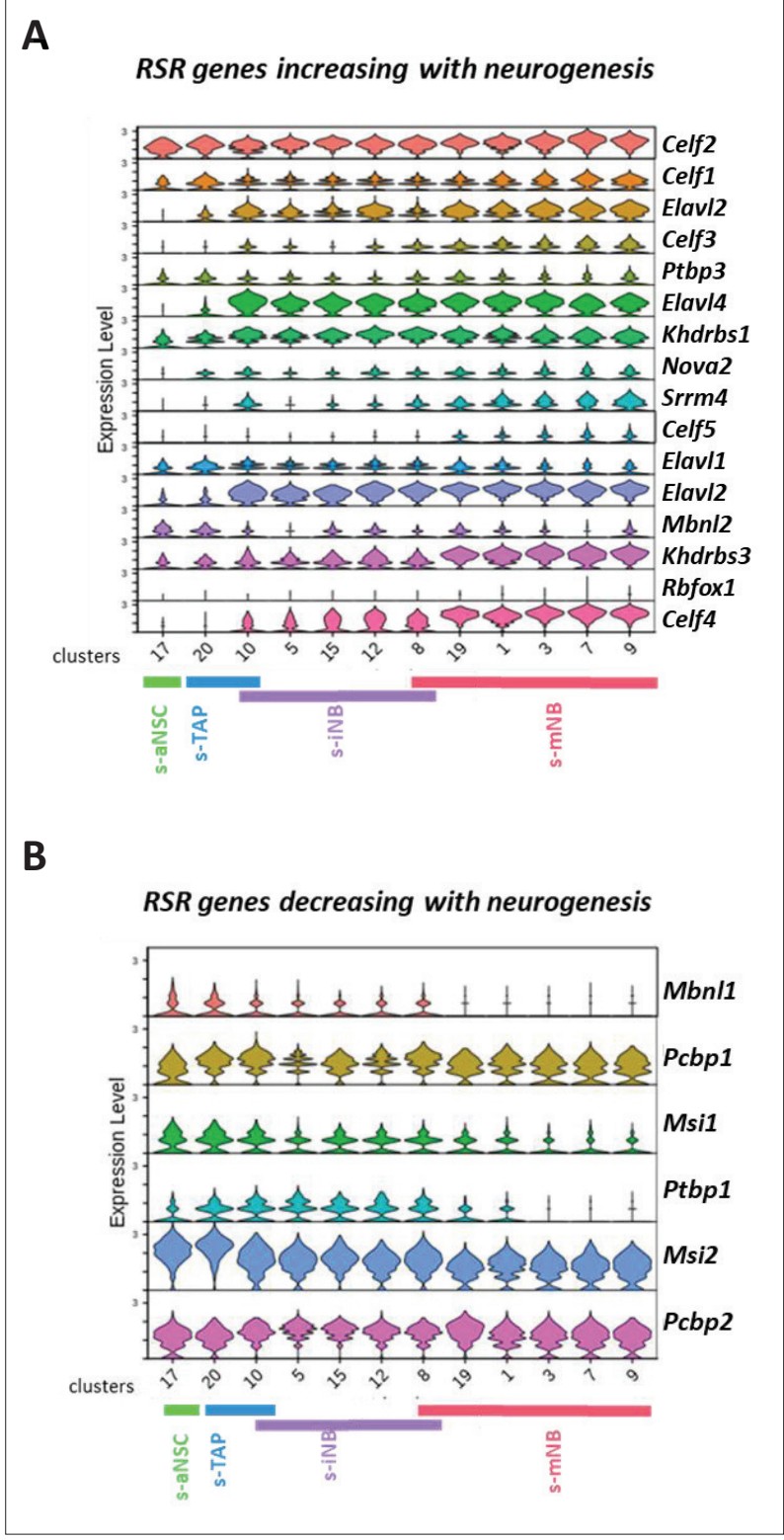

**Figure 4.** Expression of RSR genes in scRNA-seq neurogenic clusters. Violin diagrams illustrating the expression of the RSR genes increasing (**A**) or decreasing (**B**) with neurogenesis.

## The Dcx status identifies two molecularly distinct subpopulations in s-iNB

We have previously shown by immunofluorescence that not all the s-iNB were positive for doublecortin (*Dcx*) expression, a common marker of neuroblasts (*Daynac et al., 2013*). Violin plot (*Figure 3D*) and feature plot (*Figure 5A*) show that *Dcx* expression was restrained to s-mNB and to clusters related to s-iNB in which it increased consistently with their hierarchical ordering we defined above. Variations in *Dcx* expression among iNB-related clusters led us to analysze separately *Dcx^low^* and *Dcx^high^* cells in these clusters (*Figure 5—figure supplement 1*). The ratios of numbers of cells per cluster indicated a greater recovery of *Dcx^low^* cells as compared to *Dcx^high^* cells in s-iNB-related clusters from irradiated brains (*Figure 5B*), reflecting their temporal ordering.

Characterization of differentially expressed genes according to the *Dcx* status in these clusters showed 588 unique genes expressed in the *Dcx^low^* cells and 98 genes specifically expressed in the *Dcx^high^* cells. Functional enrichment profiling using the g:Profiler software illustrates biological differences between both populations: genes related to metabolism of RNA were overrepresented in the *Dcx^low^* cells, whereas genes linked to GABAergic synapse, neuronal system were rather overexpressed in the *Dcx^high^* cells (*Figure 5—source data 1*). Notably, 55% of *Dcx^high^* cell specific genes were s-mNB genes (*e.g. Robo2, Gap43, Nav3*). Further analysis of RSR genes expression in function of the *Dcx* status, clearly confirms that progressive transcriptomic changes occurred between *Dcx^low^* and *Dcx^high^* cells (*Figure 5C and D*). However, RSR gene profiles of *Dcx^low^* cells in clusters related to s-iNB were obviously different from that related to both s-TAP and s-NSC, and that of *Dcx^high^* cells from that of s-mNB. Similar results were obtained with cells from irradiated brains (*Figure 5—figure supplement 2*).

These data clearly show that the clusters related to s-iNB contained cells engaged in neuronal differentiation that are molecularly distinct from the other types of SVZ neurogenic cells already characterized This suggest that s-iNB correspond to a new stage of cycling neurogenic progenitors progressively engaged in neuronal differentiation.

We next took advantage of the DCX-CreERT2::CAG-floxed-eGFP mice model, allowing the expression of eGFP in cells expressing *Dcx* after the administration of Tamoxifen to sort eGFP$^+$CD24$^+$EGFR$^+$ cells corresponding to *Dcx^high^* iNB and eGFP$^-$CD24$^+$EGFR$^+$ cells containing the *Dcx^low^* iNB to distinguish further these two populations at the functional level. As shown in *Figure 6A and B*, recombinant eGFP$^+$ iNB exhibited a clonogenic potential and a rate of population doublings quite similar to eGFP$^-$ iNB and total iNB, contrary to eGFP$^+$ mNB, which did not show clonogenic potential and did not proliferate at long-term in vitro. Moreover, we showed that both eGFP$^+$ iNB and eGFP$^-$ iNB have the capacity to generate the three neural lineages, neurons, astrocytes and oligodendrocytes, when plated for 5–7 day in the appropriate differentiation media (*Figure 6C*), similarly as s-iNB (*Figure 1C*).

By contrast, when eGFP$^+$iNB were transplanted into the brain of C57Bl/6 J mice, no eGFP$^+$ cells persisted at long-term (5 weeks; *Table 2*), suggesting that the in vivo plasticity and regenerative properties of s-iNB was rather associated to *Dcx^low^* iNB.

## Discussion

This study bridges the cellular and molecular characterizations of SVZ neural progenitor populations. We combined the transcriptional profiling using DNA microarrays of SVZ that were sorted by an efficient FACS strategy (*Daynac et al., 2015*) and large-scale scRNAseq to decipher the progressive molecular and cellular changes involved in SVZ neurogenesis. Our model of regeneration of the neurogenic niches after brain irradiation (*Daynac et al., 2013*) allowed a unique way to determinate the sequential regeneration of the different types of neurogenic cell clusters identified by scRNAseq that were precisely characterized using the respective molecular signatures of sorted-SVZ cells, providing new insights onto the progression of SVZ neurogenesis. We show that iNB is an abundant population of cycling progenitors, which is more advanced towards neuronal differentiation than TAP, while retaining unexpected stem cell properties unlike mNB. We suggest that major splicing regulations in iNB might be critical for the final commitment to the neuronal fate.

Previous studies of the different sub-populations of SVZ progenitors were carried out using transcriptomic approaches based on the expression of various more or less specific markers. These approaches enabled the identification of quiescent and activated neural stem cells as well as mature

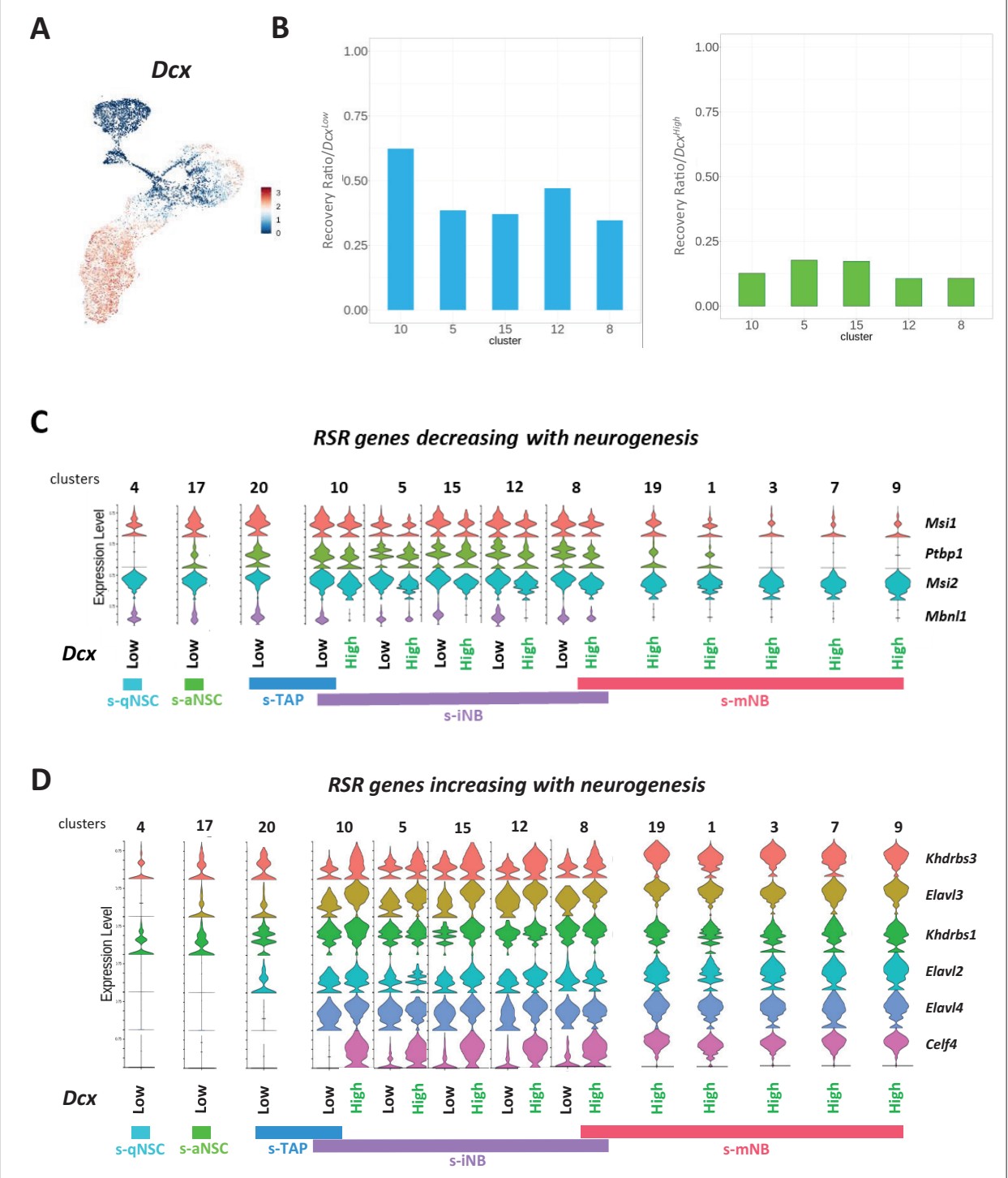

**Figure 5.** *Dcx* expression in the neurogenic cell clusters. (**A**) Feature plot. (**B**) Barplot representations of the recovery ratios of *Dcx^low^* (left) and *Dcx^High^* (right) in the iNB clusters of the irradiated brains, calculated as in **Figure 3K**. Violin diagrams illustrating the expression level of RSR genes decreasing (**C**) and increasing (**D**) with neurogenesis in *Dcx^low^* and *Dcx^High^* in neurogenic cell clusters.

The online version of this article includes the following source data and figure supplement(s) for figure 5:

**Source data 1.** Differentially expressed genes of s-INB *Dcx^High^* and s-INB *Dcx^Low^* cells and functional profiling using g:Profiler interface.

**Figure supplement 1.** Expression of *Dcx* in scRNA-seq neurogenic clusters.

**Figure supplement 2.** Expression of RSR genes in scRNA-seq neurogenic clusters in irradiated brains.

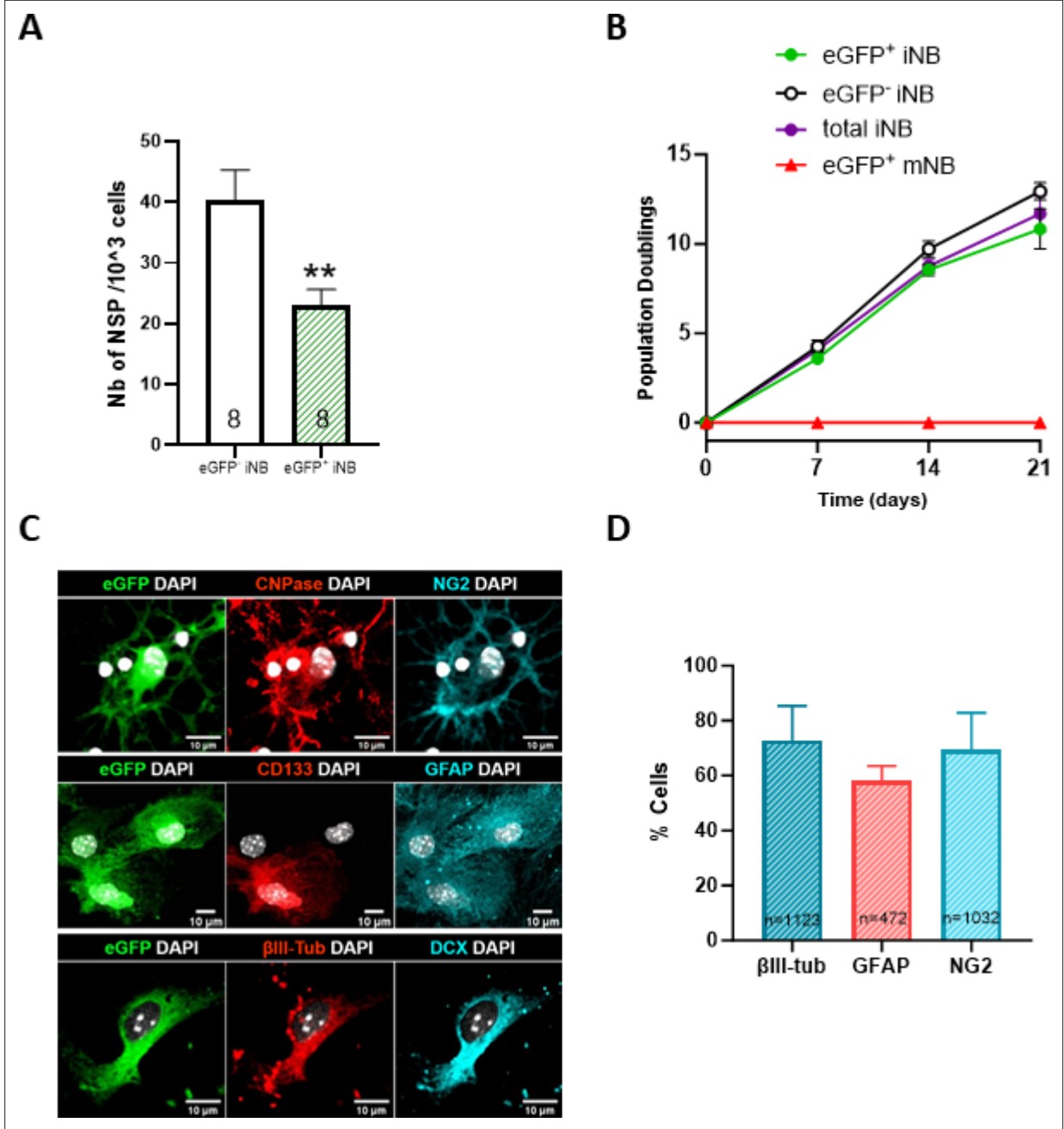

**Figure 6.** Plasticity *of Dcx*$^{High}$ iNB in vitro. (**A**) Clonogenic potential of FACS-isolated eGFP$^+$iNB and eGFP$^-$iNB, and recombined eGFP$^+$mNB isolated from DCX-CreERT2::CAG-floxed-eGFP mice, 7 days after the first injection of tamoxifen. (**B**) Population doublings (PD) of FACS-isolated eGFP$^+$iNB and eGFP$^-$iNB, eGFP$^+$mNB cells and total iNB cells (regardless of eGFP status). No statistical difference was found in the growth rate of the different isolated populationss, Mann-Whitney U-test., (n=8) (**C**) Representative images of immunofluorescence of freshly sorted eGFP$^+$iNB cells cultured in oligodendrocyte,astrocyte, or neuronal differentiation medium, and stained for NG2 and CNPase, GFAP and CD133, βIII-Tubulin and DCX, respectively. Scale: 10 µm. (**D**) Quantification of the percentage of eGFP$^+$ iNB among DAPI cells cultured in differentiation media that differentiate in either neurons (βIII-tubulin), oligodendrocytes (NG2) or astrocytes (GFAP).

neuroblasts, but were confronted with the strong influence of the cell cycle on cell clustering. Indeed, neural progenitors in these studies cycling have been gathered in either 'mitotic' clusters (*Cebrian-Silla et al., 2021*; *Zywitza et al., 2018*; *Llorens-Bobadilla et al., 2015*) or 'neural progenitor cells' clusters (*Dulken et al., 2017*) that lacked clear biological significance and prevented the identification of subtypes of SVZ cycling progenitors. Our study, combining for the first time characterization of FAC-sorted cells and an irradiation-based model of sequential regeneration, clearly distinguished the

**Table 2.** Transplantations of eGFP+s-iNB, eGFP-s-iNB and eGFP+s-mNB freshly isolated from DCX-CreERT2::CAG-floxed-eGFP mice model.

Table recapitulating the immunohistological analyses of the numbers of eGFP+ cells in different regions in recipient C57Bl6/J mice brains, 5 weeks after transplantation.

| | | | IS | | SVZ | | CC | | OB | | Total eGFP + cells |
|---|---|---|---|---|---|---|---|---|---|---|---|
| | | eGFP + engrafted cells | NbeGFP + cells | % | NbeGFP + cells | % | NbeGFP + cells | % | NbeGFP + cells | % | 0 |
| s-iNB eGFP + | Animal1 | | 0 | 0.0% | 0 | 0.0% | 0 | 0.0% | 0 | 0.0% | 0 |
| s-iNB eGFP+ | Animal2 | | 0 | 0.0% | 0 | 0.0% | 0 | 0.0% | 0 | 0.0% | 0 |
| s-iNB eGFP + / eGFP- | Animal1 | | 0 | 0.0% | 0 | 0.0% | 0 | 0.0% | 0 | 0.0% | 0 |
| s-iNB eGFP+ / eGFP- | Animal2 | | 0 | 0.0% | 0 | 0.0% | 0 | 0.0% | 0 | 0.0% | 0 |
| s-iNB eGFP +eGFP- | Animal3 | | 0 | 0.0% | 0 | 0.0% | 0 | 0.0% | 0 | 0.0% | 0 |
| s-mNB eGFP+ | Animal1 | | 0 | 0.0% | 0 | 0.0% | 0 | 0.0% | 0 | 0.0% | 0 |
| s-mNB eGFP + | Animal2 | | 0 | 0.0% | 0 | 0.0% | 0 | 0.0% | 0 | 0.0% | 0 |
| s-mNB eGFP + | Animal3 | | 0 | 0.0% | 0 | 0.0% | 0 | 0.0% | 0 | 0.0% | 0 |

molecular profiles of TAP and iNB among cycling progenitors reflecting differences in their respective potentials in vitro and in vivo.

Our results constitute thus a valuable resource for delineating the molecular transitions occurring during neurogenesis that are orchestrated by numerous differentially expressed genes and alternative splicing variants. The transition from qNSC to aNSC has been abundantly investigated in the literature (*Dulken et al., 2017*; *Llorens-Bobadilla et al., 2015*), but the transcriptional shift leading to the differentiation of neural progenitors into neuroblasts has not been thoroughly analyszed to date. We showed that it occurs in immature neuroblasts, which form a relatively abundant subpopulation of SVZ cells, which exhibit specific transcriptional features distinguishing them from TAP and mNB.

We formerly described the isolation of s-iNB as a subset of cycling SVZ cells that expressed both CD24, a neuroblast marker, and EGFR, a marker of cycling progenitors (*Daynac et al., 2015*; *Daynac et al., 2013*; *Daynac et al., 2016*). This was consistent with prior findings describing a subset of neuroblasts co-expressing Ascl1 and Dcx or CD24[low] expressing low levels of EGFR in the adult SVZ (*Pastrana et al., 2009*; *Kim et al., 2011*; *Ponti et al., 2013*). Here, we evidenced that s-iNB comprise a molecularly distinct population of cycling cells at the transition between TAP and mNB, and exhibiting molecular characteristics of both these cell types.

The capacity of neuroblasts to reorient towards the glial lineage in pathological contexts have been reported previously (*El Waly et al., 2018*; *Jablonska et al., 2010*; *Klein et al., 2020*). Consistent with the molecular findings showing that s-iNB kept transcriptional features of NSC and TAP, we showed that s-iNB also kept in vitro and in vivo a regenerative potential similar to that of s-aNSC and s-TAP. Importantly, this may also include the capacity to dedifferentiate in vivo and generate NSC as suggested by the detection of GFAP-positive engrafted cells persisting in the SVZ.

Interestingly, s-iNB corresponded to several scRNAseq clusters, which were not clearly characterized in the literature. Our model of SVZ regeneration established the hierarchical ordering of these clusters, in which we showed progressive molecular changes leading to the differentiation into migrating neuroblasts. This includes the increase in *Dcx* expression. *Dcx*[high] cells in these clusters have transcriptional features confirming they are more engaged in the differentiation than their *Dcx*[low] counterparts. Importantly, we showed that *Dcx*[high] and *Dcx*[low] cells in these clusters remained molecularly distinct from both TAP and mNB. We thus propose a revision of the current model of SVZ neurogenesis by introducing iNB cells, as highly cycling progenitors that undergo progressive neuronal differentiation but kept multipotency (*Figure 7*). iNB cells could be further divided in two subclasses of cells depending on their progression into the neuronal differentiation process: iNB1 cells, corresponding to *Dcx*[low] iNB, immediately succeeding the TAP and iNB2 cells, corresponding to

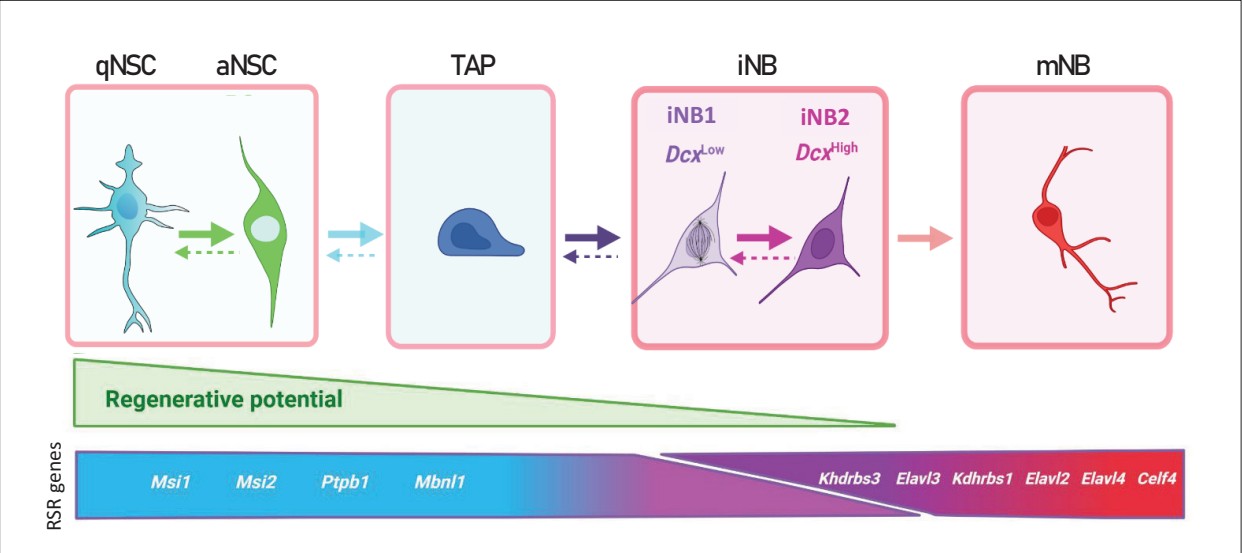

**Figure 7.** Proposed model of adult mouse SVZ neurogenesis involving major changes in RSR genes expression. Possible interconversion of iNB states would require further experimental confirmation.

*Dcx*^high^ iNB and preceding the stage mNB. Importantly, we showed that both iNB1 and iNB2 cells kept transcriptomic features that differentiated them from both TAP and mNB, whereas iNB1 cells kept a regenerative potential both in vitro and in vivo.

One of the most important findings of this study is the first characterization of AS events and RSR expression variations occurring along with neurogenesis. AS events are known to play important roles in neural cell differentiation, neuronal migration, synapse formation, and brain development (*Grabowski, 2011*; *Yano et al., 2010*; *Mauger et al., 2015*; *Shin and Kim, 2013*; *Zhou et al., 2012*). Neurons are characterized by a pervasive use of alternative exons (*Irimia et al., 2014*; *Li et al., 2015*). Consistently, we showed a dramatic increase in DSG at the transition between s-iNB and s-mNB. These alternative splicing events are clearly related to major changes in RNA splicing regulators (RSR) expressions occurring in s-iNB. Indeed, we identified RSR genes whose expression increased with neurogenesis and others whose expression decreased with neurogenesis. Consistently, some genes upregulated in s-aNSC, s-TAP, such as *Msi1* and *Msi2,* have a known role in maintenance and renewal of stem cells (*Fox et al., 2015*; *Sakakibara and Okano, 1997*; *Yano et al., 2016*), while some others upregulated in s-mNB, such as *Elavl3 and 4,* have known roles in many neuronal processes (*Yano et al., 2016*; *Bronicki and Jasmin, 2013*). Interestingly, genes of both groups of genes were overexpressed in scRNAseq clusters related to s-iNB, which thus undoubtedly distinguished these cells from both s-TAP and s-mNB and confirmed that important molecular switches for neuronal differentiation occurred in D cells.

Further analysis of scRNAseq clusters showed that these transcriptomic switches occurred sequentially from iNB1 cells to iNB2 cells. Indeed, the expression of *Elavl4* progressively increased from the *Dcx*^low^ iNB1 cells to the *Dcx*^high^ iNB2 cells fractions, confirming their progressive commitment into neuroblasts, and overall, the interest of discriminating these two cell types in the model of SVZ neurogenesis we propose (*Figure 7*). Strikingly, the splicing repressor, *Ptbp1*, was expressed in iNB1 cells at levels similar to those in s-aNSC and s-TAP, but decreased in iNB2 cells, and was undetectable in migrating neuroblasts. This suggests that expression of *Ptbp1* may play an important role in maintaining the regenerative properties in iNB1 cells, which should be further investigated.

Our study has been performed on cells collected from the SVZ the LWs of the ventricles of young adult mice. The transcriptional signatures we provide here may thus be useful for further characterization of neurogenic cells present in the septal wall of the SVZ (*Mizrak et al., 2019*).

To conclude, we have characterized the molecular features of iNB cells, a relatively abundant population in the adult SVZ that kept unexpected plasticity and multipotency that persist in the aged brain (*Daynac et al., 2015*). They may thus represent a new target in regenerative medicine. We have shown that these cells undergo major transcriptomic changes involving AS and associated to switches

in RSR genes expression. Dysregulations of many of these RSR genes have been involved in cancer and neurodegenerative diseases (*Kang et al., 2014*). Further characterization of iNB cells using the tools we developed may be useful to determine the potential therapeutic targeting of these cells in various brain pathologies.

# Materials and methods

## Key resources table

| Reagent type (species) or resource | Designation | Source or reference | Identifiers | Additional information |
|---|---|---|---|---|
| Genetic reagent (mouse) | β-actin:eGFP | The Jackson Laboratory (*Okabe et al., 1997*) | | |
| Genetic reagent (mouse) | DCX-CreERT2 ::CAG-CAT-eGFP | kindly given by Pr Couillard-Despres Sebastien (*Zhang et al., 2010*) | | |
| Antibody | PE-rat Anti-mouse monoclonal CD24 (clone 1/69) | BD Biosciences | Cat#307567; RRID:AB_2034001 | 1:50 |
| Antibody | FITC-rat Anti-mouse monoclonal CD15 (clone MMA) | BD Biosciences | Cat#340703; RRID:AB_400103 | 1:50 |
| Antibody | BV421-mouse monoclonal anti-human CD15 (Clone W6D3) | BioLegend | Cat#323039; RRID:AB_2566519 | 1:50 |
| Antibody | Alexa647-conjugated epidermal growth factor (EGF) ligand (clone 30H45L48) | Thermo Fisher Scientific | Cat#MA7-00308-A647; RRID:AB_2662334 | 1:200 |
| Antibody | Goat polyclonal anti-GFP | Abcam | Cat#ab6673; RRID:AB_305643 | 1:300 |
| Antibody | Rabbit polyclonal Anti-GFP | Abcam | Cat#ab290 | 1:300 |
| Antibody | Mouse monoclonal anti-mouse O4 (clone 81) | Merck Millipore | Cat#MAB345; RRID:AB_2313768 | 1:100 |
| Antibody | Rabbit polyclonal anti-mouse NG2 | Merck Millipore | Cat#AB5320, RRID:AB_91789 | 1:100 |
| Antibody | Goat polyclonal anti-mouse Dcx C-18 | Santa Cruz | Cat#sc-8066, RRID:AB_2088494 | 1:100 |
| Antibody | Rabbit polyclonal anti-mouse Dcx | Cell Signaling | Cat#4604, RRID:AB_561007 | 1:400 |
| Antibody | Rabbit monoclonal anti-mouse Tubulin b3 (TUBB3) | Covance | Cat#PRB-435P, RRID:AB_29163 | 1:200 |
| Antibody | Mouse monoclonal anti-mouse Gfap (clone GA5) | Merck Millipore | Cat#MAB3402, RRID:AB_94844 | 1:400 |
| Antibody | Rabbit polyclonal anti-mouse Gfap | Merck Millipore | Cat#G9269, RRID:AB_477035 | 1:400 |
| Antibody | Rat monoclonal anti-CD133 (clone 13A4) | Merck Millipore | Cat#MAB4310 | 1:100 |
| Antibody | Mouse monoclonal anti-mouse CNPase (clone 11-5B) | Merck Millipore | Cat#MAB326 | 1:100 |

*Continued on next page*

*Continued*

| Reagent type (species) or resource | Designation | Source or reference | Identifiers | Additional information |
|---|---|---|---|---|
| Antibody | Goat polyclonal anti-mouse Olig2 | Biotechne | Cat#AF2418, RRID:AB_2157554 | 1:200 |
| Antibody | Mouse monoclonal anti-mouse NeuN (clone A60) | Merck Millipore | Cat#MAB377, RRID:AB_2298772 | 1:100 |
| Antibody | Alexa Fluor 594_ donkey polyclonal anti-Mouse IgG (H+L) Secondary Antibody | Thermo Fisher Scientific | Cat#A21203, RRID:AB_141633 | 1:400 |
| Antibody | Alexa Fluor 647_ donkey polyclonal anti-mouse IgG (H+L) Secondary Antibody | Thermo Fisher Scientific | Cat#A21202A-31571, RRID:AB_162542 | 1:400 |
| Antibody | Alexa Fluor 647_ donkey polyclonal anti-mouse IgG (H+L) Secondary Antibody | Thermo Fisher Scientific | Cat#A32787, RRID:AB_2762830 | 1:400 |
| Chemical compound, drug | Percoll | Merck Millipore | Cat#9048-46-8 | 22% |
| Chemical compound, drug | Poly-L-ornithine solution | Merck Millipore | P4957 | |
| Chemical compound, drug | Poly-D-lysine solution | Merck Millipore | 25988-63-0 | |
| Chemical compound, drug | NeuroCult Basal medium | STEMCELL Technologies | Cat#05700 | |
| Chemical compound, drug | Heparin Solution | STEMCELL Technologies | Cat#07980 | |
| Chemical compound, drug | Epidermal Growth Factor Protein, human recombinant | Merck Millipore | Cat#GF144 | |
| Chemical compound, drug | Fibroblast growth Factor basic, human recombinant | Merck Millipore | Cat#GF003AF | |
| Chemical compound, drug | DMEM:F12 medium | Thermo Fisher Scientific | 31331028 | |
| Chemical compound, drug | B-27-Supplement Minus Antioyxidants (AO) | Thermo Fisher Scientific | Cat#10889038 | |
| Chemical compound, drug | Fetal Bovine serum | Thermo Fisher Scientific | Cat#10082139 | |
| Chemical compound, drug | Papain | Worthington Biochemical | Cat#LK003150 | |
| Chemical compound, drug | Tamoxifen | Sigma-Aldrich | Cat#T5648 | |
| Chemical compound, drug | D-(+)-Glucose | Merck Millipore | Cat#G7021 | |
| Chemical compound, drug | Sodium bicarbonate | Merck Millipore | Cat#S5761 | |
| Chemical compound, drug | HEPES sodium salt | Merck Millipore | Cat#H3784 | |

*Continued on next page*

*Continued*

| Reagent type (species) or resource | Designation | Source or reference | Identifiers | Additional information |
|---|---|---|---|---|
| Chemical compound, drug | Insulin from bovine pancreas | Merck Millipore | Cat#I1882 | |
| Chemical compound, drug | Apo-Transferrin human | Merck Millipore | Cat#T2252 | |
| Chemical compound, drug | Progesterone | Merck Millipore | Cat#P6149 | |
| Chemical compound, drug | Putrescine | Merck Millipore | Cat#P7505 | |
| Chemical compound, drug | Sodium selenite | Merck Millipore | Cat#S9133 | |
| Chemical compound, drug | Papain | Worthington Biochemical | Cat#LK003150 | |
| Commercial assay or kit | Red blood Cell lysis solution | Milteny Biotec | Cat#130-094-183 | |
| Commercial assay or kit | Dead Cell Removal Kit | Milteny Biotec | Cat#130-090-101 | |
| Commercial assay or kit | RNeasy Micro Kit | QIAGEN | Cat#74004 | |
| Commercial assay or kit | RNase-Free DNase Set | QIAGEN | Cat#79254 | |
| Commercial assay or kit | Agilent High Sensitivity DNA Kit | LabChip | Cat#5067–4626 | |
| Commercial assay or kit | Chromium Next GEM Chip G Single Cell Kit | 10 X GENOMICS | Cat#PN-1000127 | |
| Commercial assay or kit | Dual Index Kit Plate TT Set A | 10 X GENOMICS | Cat#PN-1000215 | |
| Software, algorithm | RStudio | RStudio | https://www.rstudio.com/ | |
| Software, algorithm | Transcriptome Analysis Console (TAC 4.0) | Thermo Fisher Scientific | | |
| Software, algorithm | limma | *Ritchie et al., 2015*; *Smyth et al., 2015* | | |
| Software, algorithm | EventPointer | *Romero et al., 2016a*; *Romero et al., 2016b* | | |
| Other | DAPI stain | Sigma-Aldrich | D1306 | Materials and methods >Immunolabeling 1 µg/ml |
| Other | Hoechst 33258 | Thermo Fisher Scientific | Cat#H3569 | Materials and methods> Cell sorting 1 µg/ml |
| Other | Propidium iodide | Sigma-Aldrich | Cat#P4170 | Materials and methods> Cell sorting 1 µg/ml |
| Other | Corn oil | Sigma-Aldrich | Cat#C8267 | Tamoxifen solution 40 mg/ml |
| Other | Solution de paraformaldéhyde, 4% en PBS | Thermo Fisher Scientific | Cat#15670799 | Materials and methods > Tissue processing |

*Continued on next page*

*Continued*

| Reagent type (species) or resource | Designation | Source or reference | Identifiers | Additional information |
|---|---|---|---|---|
| Other | Triton X-100 | Sigma-Aldrich | Cat# 11332481001 | Materials and methods >Immunolabeling 0.1% |
| Other | 70 µm cell strainer | Thermo Fisher Scientific | Cat#10788201 | Materials and methods > Single cell library preparation |
| Other | 2.5 µl, Model 62 RN SYR, Small Removable NDL, 22 s ga, 2 in, point style 3 | Hamilton | Cat#87942 | Materials and methods > Transplants |
| Other | Small Hub 33 G | Hamilton | Cat#074750 | Materials and methods > Transplants |

## Mice

All experimental procedures complied with European Directive 2010/63/EU and the French regulations. The protocols were approved by our institutional ethical committee (CEtEA 44) and authorized by the « Research, Innovation and Education Ministry » under registration number APAFIS#25978–2020061110358856 v1. Adult male C57BL/6 J were obtained from Janvier Laboratories. Adult male β–actin:eGFP (*Okabe et al., 1997*) and DCX-CreERT2::CAG-CAT-eGFP mice kindly given by Pr Couillard-Despres Sebastien (*Zhang et al., 2010*). Experiments were performed with male animals aged 8 and 10 weeks. Tamoxifen (TAM, T-5648, Sigma-Aldrich) was dissolved in corn oil (C-8267, Sigma-Aldrich) at 40 mg/ml. To analysze the expression of the *Dcx* reporter gene, 20 mg TAM/g body weight was administered twice daily by gavage over a period of 5 consecutive days.

## Cell sorting

Lateral ventricle walls of the SVZ were microdissected and dissociated as previously described (*Daynac et al., 2015*; *Daynac et al., 2013*; *Daynac et al., 2016*) with the modification that cells were centrifuged at 250 × *g* for 20 min at 4 °C without brake on a 22% Percoll gradient (GE Healthcare) to remove myelin prior to singlecell suspension labelling. The antibodies to distinct cell populations were anti-CD24 phycoerythrin [PE]-conjugated (Rat IGg2b; BD Biosciences, 1:50, RRID:AB_2034001), CD15/LeX fluorescein isothiocyanate-[FITC] conjugated (clone MMA, mouse IgM; BD Biosciences, 1:50, RRID:AB_400103), CD15/LeX- Brilliant violet-421-[BV421] (Mouse IgG1, κ; BioLegend, 1:50, RRID:AB_2566519) and Alexa647-conjugated epidermal growth factor (EGF) ligand (Clone 30H45L48, Thermo Fisher Scientific, 1:200, RRID:AB_2662334). Cells were gated following the fluorescence minus one (FMO) control. Immediately prior to FACs sorting, propidium iodide (PI) or Hoechst 33258 was added to a final concentration of 1 µg/ml to label the dead cells. Cells were sorted on an INFLUX cell sorter equipped with an 86 µm nozzle at 40 psi (BD Biosciences). All the data were analyszed using FlowJo software (Tree Star, Ashland, OR).

## Cell culture expansion and differentiation

Clonogenicity assay and population doublings: FACs-purified populations were collected in Neuro-Cult medium complemented with the proliferation supplement (STEMCELL Technologies), 2 µg/mL heparin, 20 ng/mL EGF and 10 ng/mL FGF-2, at a density of 1x10³ cells/well in 96-well tissue culture plates coated with poly-D-lysine (Merck Millipore).

Five days after plating, the neurospheres were counted to determine the clonogenicity of each cell population.

Neurospheres were then mechanically dissociated and sub-cultured in 24-well plates over 3 weeks to measure the population doublings at each passage (1 passage/week).

Differentiation assays: just after sorting, cells were placed on Poly-L-ornithine (P4957, Merck Millipore) coated 24-well culture plates at 40,000 cells/ml in DMEM:F12 with several supplementation according to the lineage: astroglial: 2% B27 MAO (minus antioxidant, Thermo Fisher Scientific) and 2% fetal bovine serum (10082139, Thermo Fisher Scientific), oligodendroglial: 10 ng/mL FGF-2, a defined hormone mix including Glucose (G-7021, Sigma-Aldrich 50 gm/l) /NaHCO₃ (S-5761, Sigma-Aldrich, 10 mg/l) / HEPES (H-3784, Sigma-Aldrich, 1.3 mM), insulin (I-1882, Sigma-Aldrich, 22.5 mg/l), Apo

transferrin (T-2252, Sigma-Aldrich, 0.09 g/l), progesterone (P-6149, Sigma-Aldrich, 18 µM), putrescine (P-7505, Sigma-Aldrich, 8.7 mg/l), and sodium selenite (S-9133, Sigma-Aldrich, 4.7 µM), and neuronal: 2% B27 MAO. Cultures in hypoxic conditions (4%O2) at 37 °C for 5 (oligodendroglial or astrocytic differentiations) or 7 days (neuronal differentiation).

## Transplants

Three injections of 1 µl of freshly sorted eGFP$^+$ cells from β−actin:eGFP or DCX-CreERT2:CAG-CAT-eGFP mice were performed in the striatum at proximity of SVZ of wild-type C57Bl6/J recipient mice using the following coordinates: (*Obernier and Alvarez-Buylla, 2019*) AP, 0.0; L, 1.4; V, –2.1; (*Codega et al., 2014*) AP, 0.5; L,1.1; V, –2.2; (*Tong et al., 2015*) AP, 1.0; L, 1.0; V, –2.5 mm relative to bregma, as described before by *Codega et al., 2014*. For CD1 mouse transplants with freshly sorted eGFP$^+$ cells from DCX-CreERT2:CAG-CAT-eGFP mice the injection coordinates were redefined from the ratio of the distance bregma to lambda for the C57Bl6/J model over the distance bregma to lambda for the CD1 model.

The transplantations were performed using a small animal stereotaxic apparatus (Kopf model 900) with a 2.5 µl Hamilton syringe (Hamilton, Bonaduz, Switzerland). Recipient mice were sacrificed 5 weeks after transplantation.

## Tissue processing, immunolabeling, and microscopy

Deeply anaesthetized animals received a transcardial perfusion of 4% paraformaldehyde. Brains were post-fixed overnight in 4% PFA and cryoprotected in 30% sucrose/PBS. Serial coronal cryostat sections were made at 14 µm of thickness (Leica CM3050S). For each brain, all the sections from the hippocampus to olfactive bulb were deposited on a set of 14 slides with a 150 µm step. Sections underwent permeabiliszation in phosphate buffered saline (PBS) containing 0.3% Triton X100 and 1% of Bbovine Sserum Aalbumin 1 hr at room temperature. Sections were incubated with primary antibodies PBS containing 0.1% Triton X100 overnight at 4 °C. Secondary antibodies used were Alexa 594, 488, and 647 and were applied at 1:400 (Thermo Fisher Scientific) for 2 hr at room temperature. Each staining was replicated in at least three slides from different mice.

Cells were fixed with 4% PFA for 15 min at room temperature and rinsed three times with PBS. Staining was performed as above except that antibodies were applied for 2 hr at room temperature.

The following antibodies used were: rabbit anti-eGFP (Abcam, ab290, 1:300), goat anti-GFP (Abcam, ab6673, 1:300), mouse anti-O4 (Millipore, MAB345, 1:100), rabbit anti-NG2 (Millipore, AB5320, 1:100), goat anti-Olig2 (RαD, AF2418, 1:200), mouse anti-NeuN (Millipore, MAB377, 1:100), rabbit anti-Dcx (Cell Signaling, 4604, 1:400), goat anti-Dcx (Santa Cruz, sc-8066, 1:100), mouse anti-GFAP (Millipore, MAB3402, 1:400), rabbit anti-Gfap (Sigma-Aldrich, G9269, 1:400), mouse anti-CNPase (Millipore, MAB326, 1:100), rabbit anti-β-III tubulin (Covance, PRB-435P, 1:200), rat anti-CD133 (Millipore, MAB4310, 1:100).

Brightfield and fluorescent images were captured through Plan Apo X20 /Numerical Aperture (NA):1.3 oil objective and Plan Apo X40 /NA:0.75 dry objective using hybrid detection technology on a laser scanning confocal (Leica Microsystems SP8), Nikon A1R confocal laser scanning microscope system attached to an inverted ECLIPSE Ti (Nikon Corp., Tokyo, Japan).

## SVZ irradiation

Animals were anesthetized with isoflurane (3% induction for 5 min) and 1.5–2% when the mice were placed in Small Animal Radiation Research Platform (SARRP, XSTRAHL, LTD Company). First, a Cone Beam Computer Scanning (CBCT) was performed to target the SVZ and to set up the treatment planning system. Mice were placed on their stomach and two fields (−45° and +45°) were used to be homogeneous on the irradiation target. The size of the irradiation field was adapted with a multivariable collimator 10*6 mm. The SVZ received a 4 Gy dose distributed 50% by each beam and the dose output was around 3.64 Gy/min depending of the size of the field. The X-ray configuration was 220 kV, 13 mA and 0.15 mm of Copper in these conditions.

## Single-cell library preparation for single-cell RNA sequencing

SVZ lateral ventricle walls were dissected from two non-irradiated mice brains (2-month-old) and two 5 days post-irradiation 4Gy-irradiated mice brains (2-month-old).

Protocol of SVZ digestion with Papain (Worthington, LK003150) to obtain single-cell suspensions were strictly identical to those performed in the whole genome microarray experiment described above, including the Percoll gradient step to remove myelin. Cells were resuspended in 0.04% BSA in PBS passed through a 70 µm cell strainer (Thermo Fisher Scientific).

Blood cells were lysed using the Red blood Cell lysis solution (Milteny Biotec, 130-094-183) and dead cell Removal using the Dead Cell Removal Kit (Miteny Biotec, 130-090-101) were performed following the manufacter's instructions.

Single-cell suspensions of two replicates of each sample (unirradiated, 4 Gy) were adjusted to 2000 cells/µl and 10,000 non-irradiated cells and 6000 irradiated cells were loaded per channel onto Chromium Next GEM Chip Single Cell kit (ref: 1000127). Library preparation was performed using the Dual Index Kit TT (ref: 1000215) according to manufacturer's recommendations (10 x Genomics, Pleasanton, CA). Quality and quantification of libraries were done using the High Sensitivity DNA LabChip kit (ref: 5067–4626). Sequencing NGS Standard Illumina was carried out with the NovaSeq6000-S1 flow cell (ICM platform, Paris).

## Whole-genome microarrays

mRNA were isolated with the RNeasy Micro Kit with DNase treatment (QIAGEN) and sample quality was controlled using the Agilent Bioanalyser. MTA-1.0 arrays (Clariom D Affymetrix technology) were performed according to the manufacturer's protocol (Thermo Fisher). PCA and pseudotime ordering was performed using TSCAN online user interface (*Ji and Ji, 2016*). The data were analyszed with the freeware software Transcriptome Analysis Console (TAC) applying the RMA algorithm normalization. Functional profiling were generated using REACTOME and KEGG datasets obtained from the g:Profiler interface (https://biit.cs.ut.ee/gprofiler/gost).

Signatures and unique mRNA splicing isorforms of cell sorted populations were delineated using Venn diagrams comparing the five populations of neural progenitors (s-qNSC, s-aNSC, s-TAP, s-iNB, and s-mNB; https://bioinformatics.psb.ugent.be/cgibin/liste/Venn/calculate_venn.htpl).

sc-RNAseq and microarray dataset comparisons:

Published genelists were matched with microarray data raw data using aggregated Z-scores. For each gene in a genelist, gene expression amongst all samples was normalized between 0 and 1, then mean expression among samples was calculated for each population. Plots represent the distribution of mean normalized gene expression for all genes in the genelist.

## Single-cell RNA control quality and analysis

ScRNA-seq data were aligned to the GRCm38 - mm10 reference genome (cellranger-6.1.2). Data were analysed in R (version 4.1.0) using Seurat (v4.1.1). This work benefited from equipment and services from the iGenSeq core facility, at ICM.

Quality control removed outlier cells with fewer than 1200–1,500 genes per cell (nFeatures_RNA), 2350–2,700 UMI (nCount_RNA) and cells less than 10% mitochondrial content. Gene expression was normalized using the regularized negative binomial regression to normalize UMI count data (SCTransform) using 4000 variable features and the different samples (controls and irradiated). The features to use in the downstream integration procedure were determined using SelectIntegrationFeatures. After a pre-computed AnchorSet (FindIntergrationAnchors), the Seurat object were integrated using IntegrateData with the normalization method SCT. CellCycleScoring was applied to calculate the S.Score, G2M.Score and Phase. RunUMAP runs the Uniform Manifold Approximation and Projection (UMAP) dimensional reduction technique using 50 PCs. FindNeighbors computes the k.param nearest neighbors using dims = 1:50. Clusters of cells were identified by a shared nearest neighbor (SNN) modularity optimization bases clustering algorithm using FindClusters with resolutions from 0.2 to 2. The resolution 1.2 that segregates cells into 33 clusters was chosen based on Clustree graph showing the relashionship between clusterings at the different resolutions. UMAP plots were generated with UMAPPlot with default parameters.

The PrepSCTFindMarkers function was used to normalize gene expression for differential gene expression analysis among clusters, which was performed using FindAllMarkers on the SCT assay, with significance determined using a Wilcoxon rank sum test (p_val_adj <0.05) and log(Fold-Change) threshold of 0.1. A histogram representation of *Dcx* (ENSMUSG00000031285) expression in the clusters corresponding to astrocytes and neural progenitors allowed us to place a threshold expression

at 1. Cells with an expression lower than or equal to 1 correspond to $Dcx^{Low}$ cells and cells with an expression strictly higher than 1 to $Dcx^{High}$ cells. Then, we reapplied PrepSCTFindMarkers and FindAllMarkers to find differentially expressed genes between $Dcx^{Low}$ and $Dcx^{High}$ iNB.

### Gene expression score

We used AddModuleScore_Ucell (Package *UCell* version 1.3.1) to score cells based on the expression of predefined specific gene lists of sorted s-qNSC, s-aNSC, s-TAP, s-iNB and s-mNB determined by microarrays using the Mann-Whitney U statistic (*Andreatta and Carmona, 2021*). Each cell was then assigned an identity corresponding to the highest score amongst the signatures tested. Repartition of classes among clusters was visualized using the Circlize package [58](*Gu et al., 2014*).

For astrocyte and B-cell classification, we used the lists of Astrocyte and B-cell markers provided by *Cebrian-Silla et al., 2021* to score cells from the clusters 4 and 13 between 0 (min) and 1 (max). Cells were assigned to the highest score amongst both signatures.

Gene enrichment analysis:

Comparisons of the specific gene lists of $Dcx^{High}$ s-iNB and $Dcx^{Low}$ s-iNB were performed with Biological pathways using the g:Profiler interface (https://biit.cs.ut.ee/gprofiler/gost).

### Quantification and statistical analysis

Non-parametric Mann-Whitney test was conducted using Prism 8.1.2 (GraphPad Software Inc, La Jolla, CA, RRID:SCR_002798). The statistical significance was set at $p < 0.05$. The data are expressed as the mean ± SEM.

## Acknowledgements

We thank C Devanand and C Duwat for their contribution for animal experiments and G Piton for 10 x single cell library preparations. We are grateful to Marie-Justine Guerquin and Pierre Fouchet, and the ARTBIO platform (Université Paris 6) for helpful discussions and assistance in bioinformatics analyses. BC has a fellowship from ARSEP (fondation pour l'Aide à la Recherche sur la Sclérose En Plaques). This work was supported by grants of IRBIO (Commissariat à l'Energie Atomique et aux Energies Alternatives, CEA) and Electricité de France (EDF). The funders have no role in the study design, data collection and analysis, decision to publish, or preparation of the manuscript.

## Additional information

### Funding

| Funder | Grant reference number | Author |
|---|---|---|
| Fondation pour l'Aide à la Recherche sur la Sclérose en Plaques | ARSEP - 1250 | Corentin Bernou |
| Électricité de France | NeuradTox_2022 | François Dominique Boussin |
| Commissariat à l'Énergie Atomique et aux Énergies Alternatives | IRBIO | François Dominique Boussin |

The funders had no role in study design, data collection and interpretation, or the decision to submit the work for publication.

### Author contributions

Corentin Bernou, Conceptualization, Data curation, Funding acquisition, Investigation, Methodology, Writing – original draft; Marc-André Mouthon, Data curation, Validation, Investigation, Methodology, Writing – review and editing; Mathieu Daynac, Conceptualization, Methodology; Thierry Kortulewski, Benjamin Demaille, Data curation, Formal analysis, Methodology; Vilma Barroca, Data curation, Investigation; Sebastien Couillard-Despres, Nathalie Dechamps, Véronique Ménard, Methodology; Léa Bellenger, Data curation, Formal analysis; Christophe Antoniewski, Data curation, Writing – review

and editing; Alexandra Déborah Chicheportiche, Conceptualization, Data curation, Software, Formal analysis, Supervision, Validation, Methodology, Writing – original draft, Writing – review and editing; François Dominique Boussin, Conceptualization, Resources, Data curation, Formal analysis, Supervision, Funding acquisition, Investigation, Methodology, Writing – original draft, Writing – review and editing

## Author ORCIDs

Alexandra Déborah Chicheportiche ⓘ https://orcid.org/0000-0002-1102-955X
François Dominique Boussin ⓘ https://orcid.org/0000-0003-3778-4403

## Ethics

All experimental procedures complied with European Directive 2010/63/EU and the French regulations. The protocols were approved by our institutional ethical committee (CEtEA 44) and authorized by the Research, Innovation and Education Ministry under registration number APAF-IS#25978-2020061110358856 v1.

Reviewer #1 (Public Review): https://doi.org/10.7554/eLife.87083.3.sa1
Reviewer #3 (Public Review): https://doi.org/10.7554/eLife.87083.3.sa2
Author response https://doi.org/10.7554/eLife.87083.3.sa3

# Additional files

## Supplementary files

• MDAR checklist

## Data availability

The datasets generated for this article were deposited online on the ANNOTARE database (E-MTAB-12265, E-MTAB-12495). Scripts used for analysis and figures are available on GitHub (copy archived at *Chicheportiche, 2024*). Molecular signature matching analyses of the neural cells sorted in our study were performed using the datasets from: *Zywitza et al., 2018*: (List of Significantly Upregulated Genes in Subclusters related to Figures 3 and S4, TableS3, File:1-s2.0-S2211124718317327-mmc4.xlsx), *Cebrian-Silla et al., 2021*: (B cells vs. Astrocytes: DE genes and GO terms, Supplementary file 1 and scRNAseq datasets from the University of California Santa Cruz Cell Browser, https://svzneuro-geniclineage.cells.ucsc.edu), *Llorens-Bobadilla et al., 2015*: (List of cluster gene sets and GO terms, File:1-s2.0-S193459091500301X-mmc3.xls).

The following datasets were generated:

| Author(s) | Year | Dataset title | Dataset URL | Database and Identifier |
|---|---|---|---|---|
| Chicheportiche A | 2023 | Adult mouse SVZ niche at the single cell resolution | https://www.ebi.ac.uk/biostudies/arrayexpress/studies/E-MTAB-12495 | ArrayExpress, E-MTAB-12495 |
| Chicheportiche A | 2022 | Microarrays of flow cytometry sorted neural progenitor populations from SVZ of adult mouse brains | https://www.ebi.ac.uk/biostudies/arrayexpress/studies/E-MTAB-12265 | ArrayExpress, E-MTAB-12265 |

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
