## [Editor Report · eLife assessment]

This **useful** manuscript presents an intriguing potential refinement of models for adult SVZ neurogenesis, and highlights the role of RNA splicing at specific stages in the lineage. Unfortunately, the evidence does not fully support the claims, leaving it currently **incomplete**. The proposed role of RNA splicing in neuronal differentiation, though interesting, remains unexplored and would benefit significantly from targeted gene manipulation studies.

---

## [Referee Report · Reviewer #1 (Public Review)]

In this study, the authors use prospective sorting and microarray analyses, extended by single-cell RNA sequencing, in the neural stem cell niche of the subventricular zone (SVZ) to identify and refine a series of states along the continuum from quiescent neural stem cells to mature progeny. Of note, changes in the levels and subgroups of RNA splicing regulators are detailed across this continuum. Using in vitro proliferation and differentiation assays, coupled with in vivo engraftment of some prospectively sorted subsets, the authors argue that a stage they define as immature neuroblasts (iNBs) retain proliferative and multilineage differentiation capacity that is not seen in the mature neuroblast population, and is unexpected based on prior models for lineage progression in this system. This iNB stage is accompanied by a change in RNA splicing regulator expression, which is of interest due to the emerging roles for RNA processing and preferential translation within this niche.

The central tension driving the discussion between authors and reviewers, in my view, is what is required to define cells as a "molecularly distinct population" in a lineage. Is it transcript expression, in vitro potential, or more? The authors argue that sorted immature neuroblasts are a defined, separate step in the neurogenic lineage. An alternative possibility is that this population is simply cycling transit-amplifying progenitors that have initiated a transcriptional program associated with neuroblast fates - that these cells are an intermediate point on a continuum between stem cells, transit-amplifying progeny, and commitment to a neuronal (or other) fate. Despite some additions in response to initial reviews, it is still the case that much of the data presented would be equally or more effective in supporting the second model. For example, the differentially spliced gene sets in Figure S4, which are put forward by the authors to support a different molecular identity for immature neuroblasts, show that the terms enriched for immature neuroblasts are largely also found in transit amplifying progenitors (generation of neurons, neurogenesis, cell projection organization, neuron development) and/or mature neuroblasts (cell projection organization, generation of neurons), suggesting that "immature neuroblasts" are transiting between these two states and that one of their most relevant features is that they are still cycling.

These data complement several additional sc-RNAseq studies of this stem cell niche, and use a different, but similar, sorting strategy to isolate and profile subpopulations of stem/progenitor cells and neuroblast progeny. The claim that immature neuroblasts retain multipotency - the ability to generate glia and neurons - is surprising and somewhat controversial given that this has largely not been reported before under homeostatic conditions. Some factors to consider when interpreting these data are that the "immature neuroblast" populations are studied in some experiments using a transcriptional signature and a functional assay, namely the timing of reappearance of these cells after use of agents that kill rapidly dividing cells (in this case, radiation), leading to reconstitution of the lineage by previously quiescent stem cells. In a separate set of experiments, a tamoxifen-inducible labeling system is used in combination with cell-surface markers to prospectively isolate and study the differentiation potential of neuroblast populations that are assumed to be equivalent to those found in transcriptional experiments. It would be of interest in the future to confirm that the exact sorted populations (using CD24/EGFR/DCX-CreERT2::CAG) have the same transcriptional profile as those studied in earlier experiments within the paper and to confirm the purity of the sorted populations. Finally, while use is made of engraftment of sorted populations to study the differentiation and lineage potential of these immature neuroblasts, a remaining question is the relative abundance of each lineage (neurons/astrocytes/oligodendrocytes) produced by the engrafted cells - is production of glia rare, or common? Could this be due to factors such as alteration of lineage potential due to culture conditions, a disconnect between transcript expression and protein expression, or an incompletely purified starter population?

Overall, this manuscript presents an intriguing possible refinement of models for SVZ neurogenesis, and highlights the role of RNA splicing at specific stages in the lineage. It will be of interest to see if additional groups confirm these findings and whether multiplexed immunostaining, highly multiplexed flow cytometry, or other approaches focused at the proteomic level extend these findings, particularly given recent data in the developing brain that suggest transcript and protein levels are relatively poorly correlated in stem/progenitor populations.

---

## [Referee Report · Reviewer #3 (Public Review)]

Summary:

Bernou et al. propose the existence of a distinct neuroblast population with increased regenerative and differentiation potential. Their claims are based on the analysis of a sorted population identified as LeX-EGFR+CD24low, which they refer to as "immature NeuroBlasts, iNB". This population is defined by transcriptomics features that have been assessed through bulk microarray studies of sorted cells and single cell RNA sequencing of the whole SVZ- lineage. Analysis of these data sets leads to the identification of these iNBs as cycling cells with a specific expression pattern of RNA splicing machinery components. On these grounds, they propose that RNA splicing plays a key role in neuronal differentiation. Although the authors bring an innovative point to the table, their claims are not fully supported by their results.

Strengths:

Interesting Hypothesis

Weaknesses:

The comparison of their microarray data to published single-cell RNA sequencing datasets (scRNAseq) highlights the cycling nature of the iNB population. Moreover, their own cell cycle analysis on their scRNAseq data attributes G2M/S-phase stages to clusters classified as iNBs, while clusters identified as TAPs are assigned to a restricted G1/S-phase stage. However, it would be expected that TAPs, as cycling progenitors, would go through all cell cycle stages and not just the beginning of it. Thus, authors should consider the possibility that their iNB population entails a major fraction of transit amplifying progenitors (TAP) and a couple neuroblasts, as described in numerous previous studies.

Authors regard the iNB population as neuroblasts due to the capacity of their sorted population to proliferate and differentiate into diverse neural cell types (neurons, oligodendrocytes and astrocytes) in vitro. It cannot be discarded that the sorted population (LeX-EGFR+CD24low) may not be pure and may be composed of a mixture of cells in different stages, including TAPs. Such a mixture of different cell types is unavoidable in sorted populations analyzed as bulk and is precisely one of the issues solved by single cell transcriptomics. Thus, the analysis of single cells resolves transition states at higher resolution and should be preferred over bulk analysis to prevent biases in analysis.

To align the authors' findings with the existing body of literature and earlier characterizations of the SVZ niche, it is advisable to combine their single-cell RNA sequencing data with datasets that have already been published. Such integration will enable precise understanding of the identity of their iNB cells.

On another note, the role of RNA splicing on neurogenesis lacks experimental validation. Unless manipulation of RNA splicing factors is conducted, the key role of this machinery in adult neurogenesis cannot be claimed.

---

## [Author Response]

The following is the authors’ response to the original reviews.

**Reviewer 1**
Major points:R1C1: I appreciate that the data are aligned, in some points, with related studies of this niche. However, it would help the reader to have this alignment explored more extensively in the Discussion as well.

Answer: We acknowledge that the discussion would benefit from additional comparisons to the available datasets. We thus add the following comment after the first paragraph of the discussion: “Previous studies of the different sub-populations of SVZ progenitors were carried out using transcriptomic approaches based on the expression of various more or less specific markers. These approaches have made it possible to identify quiescent and activated neural stem cells as well as mature neuroblasts, but have been faced with the strong influence of the cell cycle on cell clustering. Indeed, neural progenitors in these studies cycling have been gathered in either “mitotic” clusters (Llorens et al. 2015, Zywitza et al. 2018, Cebrian et al. 2021) or “neural progenitor cells” clusters (Dulken et al. 2017) that had no clear biological significance and hindering identification of subtypes of SVZ cycling progenitors. Our study, combining, for the first time, characterization of Facs-isolated cells and an irradiation-based model of sequential regeneration, allowed to clearly distinguish the molecular profiles of TAP and iNB among cycling progenitors reflecting differences in their in vitro and in vivo respective potentials”.

R1C2: The data on multilineage differentiation, both in culture and upon engraftment, would be greatly strengthened by quantification. What is the relative yield of TUJ1/DCX-positive cells versus the other marker combinations? Specifically regarding the multilineage differentiation in vitro - because different media conditions are used to generate each lineage, it may be difficult to determine relative yield. Could a differentiation system that allows production of all 3 lineages be used instead?If the fraction of non-DCX/TUJ1-labeled progeny is low, particularly in vivo, this might suggest that while multilineage differentiation is possible, it is a much less likely cellular state outcome than production of mature neuroblasts. Some suggested references with examples of the culture conditions, experimental conditions, and discussions highlighted in the public review:Culture conditions that allow simultaneous trilineage differentiation. PMID: 17615304 Influence of culture conditions on potency: similar to issues covered in PMID: 21549325.

Answer: We agree with the reviewer that quantification of a multilineage differentiation in vitro would improve the characterization of the relative potencies of the different SVZ progenitor.

According to PMID: 17615304 and PMID: 21549325, and in agreement with our own experience, the only culture condition that allows neurosphere-derived neural progenitors to differentiate in vitro into the three lineages is the removal of mitogens from the culture medium. However, this does not work on freshly isolated SVZ cells, which remain in an undifferentiated state in this condition.

This is why we chose to use specific differentiation media for each of the 3 lineages as in Figure 1C. It is also for this reason that we performed as many experiments as possible in vivo rather than in vitro as in Figure S2. In the new version, we have added a quantitative analysis of stainings by antibodies against GFAP, CNPase or DCX of GFP-positive cells persisting at IS, where high number of grafted cells were found in Figure S2B. This was performed by using the NIS software measuring eGFP-, GFAP-, CNPase- and DCX-positive areas. The intersection between each marker and eGFP areas was then determined as a percentage of staining (Figure S2C). The results showed that approximately one third of GFP+ cells expressed GFAP or DCX. The quantitative analysis of CNPase expression was complicated by CNPase-positive host cells, but the stronger CNPase staining in eGFP-positive areas clearly revealed the expression of CNPase by a significant proportion of eGFP-positive cells.

R1C3: Additionally, for claims similar to what is currently made in the text, it would be extremely valuable to confirm the purity of the sort for each population - for example by fixing and staining the sorted fraction with additional antibodies that confirm cell identity.

Answer: We have previously shown in Daynac et al. 2013 that s-iNB expressed the neuroblast markers CD24 and DCX, but also markers of neural progenitors such as Mash1, a basic helix-loop-helix transcription factor. As suggested by the reviewer, we have further investigated the expression of other markers of neural progenitors by sorted cells. The results showed that the proportion of DLX2+ cells a marker of proliferating progenitors (Doetsch et al. 2002) was very high in aNSC/TAP (98%) and progressively decreased in iNB (82%) and mNB (25%). Similarly, the expression of the transcription factor SOX2 that plays an essential role in the maintenance of neural progenitors (PMID: 25126380) accounted for 78% of aNSC/TAP, 70% of iNB and 17% of mNB.

Altogether, these new data confirmed the identity of the different cell populations and particularly that of iNB. They are commented at the beginning of the Results and shown in Figure S1.

R1C4: Line 125: GFAP alone doesn't necessarily indicate a "conversion to NSCs" - this conclusion could be greatly strengthened by inclusion of more markers, particularly at the protein level, or cyto-architectural studies.

Answer: We agree with the reviewer that GFAP expression alone is not sufficient to evidence the presence of NSC in the SVZ. We have thus modified the text accordingly: “Importantly, eGFP+ cells were present in the SVZ of all the animals transplanted with eGFP+s-iNB and eGFP+s-NSC/TAP (Fig. 1Db, Fig. 1Dc), some of them expressing GFAP indicating the generation of astrocytes, and therefore possibly NSC”.

R1C5: Could these cellular states be reflective of preferential translation of DCX? It would be very helpful to see the flow cytometry sort data for iNBs / mNBs used in Figure 6, particularly if these cells were also fixed and stained directly for DCX protein.

Answer: As suggested by the reviewer, freshly FAC-sorted iNB and mNB were fixed and labelled with an anti-DCX monoclonal antibody after permeabilization. As shown in the figure below, we found a higher level of DCX expression in mNB than in iNB. Therefore, this result tends to indicate that the proliferation capacity is somehow related to the level of DCX expression. However, because of the relatively low importance of this result, we decided not to include them in the manuscript.

**Author response image 1. sa3fig1:** Modal histogram representation of DCX expression level in unstained, iNB and mNB cells determined by flow cytometry (FlowJo).

<R1C6: Figure S8 is all zeroes, showing the GFP+Dcxhigh NBs do not retain proliferative capacity. But we don't get a direct experimental comparison to EGFPnegative/lowDcxlow iNB engraftment, which would strengthen the conclusions of the paper.

Answer: Unfortunately, there is no method available to analyse the eGFPnegative/lowDcxlow iNB engraftment: by definition, these cells do not express eGFP and the use of a tracker is not appropriate for long periods of time — and thus a high number of cell divisions — after engraftment. However, to us, this control is not needed to conclude that GFP+Dcxhigh iNB have no (or at least a lower) stem cell potential in vivo considering that we have shown in Figure 1 and Table 1 that the whole iNB population is able to generate the different types of neural cells.

R1C7: Transplant data in Table 1 - a relatively small proportion of transplant derived cells are in OB, etc. Given that A cells are thought to cycle at least once in vivo, is this expected?

Answer: The reviewer is right considering that a relatively small proportion of transplant derived cells were found in the OB. However, we should consider that we used immunocompetent mice as receivers, which could have significantly reduced the engraftment efficiency, and the migration of engrafted cells outside the injection site.

R1C8: A caveat is that there is not much functional testing of the proposed model, especially for the interconversion of iNB states suggested by the diagram in Figure 7. The text is relatively restrained in proposing this model, so it is reasonable to keep - but perhaps should be noted that this part of the model will need additional testing.

Answer: Data presented in Figure 6 clearly suggest that Dcxhigh iNB have similar in vitro potential than Dcxlow iNB, whereas they don’t have such potential in vivo (Figure S10). This suggests that, providing they are in appropriate conditions, Dcxhigh iNB could reacquire stem/progenitor properties. However, we agree that this hypothesis requires further investigation. Therefore, as suggested by the reviewer, we have added in the Figure 7 legend: “Possible interconversion of iNB states would require further experimental confirmation.”

Additional minor points:R1C9: Introduction: the SVZ is described as "the lateral wall" - however, several works in the mouse have also examined the medial wall and callosal roof, as cited later in the intro. Suggest rephrasing the second sentence (line 48) and later sentence (line 66) to clarify that "the SVZ" encompasses all of these subregions, they are not necessarily separate niches.Answer: As indicated by the reviewer, the SVZ encompasses distinct subdomains, with NSCs having a regional identity based on their location in the lateral or septal wall of the ventricle and generating different types of neuronal and glial progeny (PMID:34259628.). To address the reviewer concern about possible confusion and clearly indicate that SVZ encompass several subdomains, we have modified the sentence line 66 as follows: “Since then, the single cell RNA-sequencing has revolutionized the field and has made it possible to precisely elucidate the transcriptome of SVZ cells present in the LW and in the septal wall which also harbors NSC niches”.

However, we did not modify the line 48, since in this sentence we just indicate that the largest neurogenic niche in the adult brain reside in the LW of the SVZ.

R1C10: Line 77: "exposure" not "exposition"

Answer: The error has been corrected in the revised manuscript.

R1C11: As noted in the Public Review - the use of the term "D1/D2" cells seems likely to confuse readers who are also versed in dentate gyrus neurogenesis. Recommend removing this term from the manuscript.

Answer: We agree that the D1/D2 terminology could bring confusion, D cells referring to Tanycytes in the hypothalamus. We now refer to iNB1 for DcxLow iNB and iNB2 for DcxHigh iNB in the revised manuscript.

**Reviewer 2**
Major comments:Lack of rigorR2C1: There is a lack of appropriate normalization controls for the microarray data. As there is a decreased level of transcription in quiescent NSCs, there needs to be a cell number control (spike-ins based on cell numbers). Without this normalization, the readout can be greatly skewed.

Answer: We agree that qNSC are marked by a decreased level of transcription due to quiescence. To overcome this problem in the Clariom assays, we thus chose to calibrate each population, with a fixed amount of cRNA and cDNA using Hela cells as internal control. We totally agree that this method is not optimal but it appears to be efficient in the end. Indeed, it should be noticed that it has been adopted, thus with the same rigor, in other microarray studies published in the field (PMID: 24811379) and also on skeletal muscle cells (PMID: 29273087). Moreover, interestingly the transcriptomic signature of qNSC matches perfectly with those from other studies and particularly to those of related clusters in single cell experiments (including ours, Figure S5). This is probably linked to the fact that more importantly that the number of cells, the main characteristic of these cells is the lack of expression of genes involved in cell proliferation and metabolism. Whatever so, these data confirming previously published are not the main information of our manuscript, which is mainly dedicated to the characterization of proliferating cells, which is not impaired by our choices of normalization.

R2C2: The absolute segregation of clusters in the single-cell analysis is currently entirely in agreement with the cell cycle stage. This suggests that in the author's analysis, the clustering in 3F is entirely shaped by the cell cycle, making that the defining characteristic of the author's definitions for their cell types. Has an analysis been done that regresses out cell cycle-associated genes to see if there are clusters for different cell states/types that are identified in the absence of cell cycle stage being the defining factor? (Barron and Li, 2016). For example, just as you would see a difference in cluster if you are a quiescent or activated NSC as compared to a neuroblast for example, even without the contribution of cell cycle. These are different cell types.

Answer: We agree that cell cycle regression would theoretically allow for further discrimination between cycling cells along successive neurogenic stages. We have already performed regression using several methods, including regressing using S- and G2/M-score regression as indicated in the Seurat workflow, removing cell cycle-related PCs from UMAP calculation as used in the Cebrian-Sylla study, and using alternative gene sets such as the ones provided by the tricycle method (PMID: 35101061). These regression methods have all been used on our datasets, the original Cebrian-Sylla datasets and a combination of our datasets with the Cebrian-Sylla original datasets to increase cell number and clustering resolution. However, none of these methods modified the clustering of cycling cells.

In fact, the strong influence of the cell cycle over clustering highlights the relevance of our depletion/replenishment approaches to decipher the molecular changes masked by the cell cycle, as discussed below.

R2C3: The use of the DCX-CreERT2 line is a lineage tracing line. Once DCX is expressed, Cre recombines the DNA to allow for fluorescence. It is binary, on or off associated with DCX expression. And once on, it is always on, whether the cell is currently expressing DCX or not. As the authors had previously described a DCXlow condition, the eGFP- cells would not reflect DCXlow, but no DCX at all. And the eGFP+ cells may not be currently expressing DCX anymore. The authors should have used a system where the DCX promoter itself drives fluorescence.

Answer: We took advantage of the DCX-CreERT2 line to demonstrate that some neural cells that have recently acquired DCX expression (i.e. eGFP+ iNB) could keep (or recover) the potential of neural progenitors in vitro. Of course, some of these GFP+ cells could have stopped to express DCX. This is probably the case when they differentiate into astrocytes and oligodendrocytes in vitro as shown in Figure 6.

Whatever so, the use of the Dcx promoter as a direct driver of eGFP fluorescence would have totally impeded our capacity to demonstrate such changes in cell fate in vivo because of the impossibility to track oligodendrocytes or astrocytes derived from iNB because of the loss of Dcx expression.

R2C4: The lack of analysis of images (differentiation, for example) limits the conclusions of the in-vitro data, and the images with unclear staining, limit the conclusions of the in-vivo experiments.

Answer: This comment is similar to that of R1C2. We have now added a quantification in Figure S2.

R2C5: The cited difference in splicing differences in cell types was interesting (though did not show up in the transcriptome enrichment analyses Fig S2) and would be something to further pursue, however, this was a very limited analysis. There was no further study of these splicing mediators beyond single-cell data.

Answer: We now show enrichments of GO terms corresponding to mRNA splicing isoforms in the different types of sorted SVZ cells (Figure S4). This analysis clearly revealed that spliced genes in SVZ cells are mainly involved in neuron development and neurogenesis. Interestingly this also showed that qNSC logically differed from the other cell types by splicing concerning genes involved in mitosis and cell cycle, consistently with their quiescent state. More importantly, GO annotations of differentially spliced isoforms further confirmed that s-TAP and s-iNB have distinct features. We agree with the reviewer that further analysis of splicing mediators would be very important for understanding molecular changes involved in neurogenesis. However, we think that it is largely beyond the scope of this study.

R2C6: Fig 1C - Show values, not just pictures. You may need to shift your current differentiation paradigm to do so by removing growth factors instead of unique differentiation conditions.

Answer: See the answer to R1C2.

R2C7: Fig S1A - Stainings for GFAP and DCX are not clear. It is very hard to distinguish which cells are associated with these signals.

Answer: This figure (now Figure S2A) shows an eGFP+iNB cell (white arrow) that has reached the rostral migratory stream and expressed DCX (inset a3), but not GFAP (inset a2). This is now indicated in the figure legend. We have also moved the arrow for more clarity.

R2C8: Fig S1B2 - There is red staining everywhere, so it is very hard to see a specific CNPase signal.

Answer: We have added a new figure (Fig S2B) distinguishing eGFP+CNPase+ cells (yellow arrows) from eGFP+CNPase- cells (white arrow).

R2C9: Line 174 - It's the mRNA that you are detecting is being downregulated - be more specific as you are not showing protein downregulation.

Answer: We specified, "encoding" a major splicing repressor in the Line 174 text to refer to the mRNA: “Interestingly, Ptbp1, encoding a major splicing repressor”.

R2C10: Line 189 - text in this line have some clusters not shown in the figure - (clusters 6 and 15, DCX+ Ki67+ neuroblasts) - which would be an important thing to visualize. As is shown now, the authors are only showing that iNBs are similar to mitotic TAPs.

Answer: Clusters 6 and 15 have been added to Figure S5.

R2C11: Fig 3D-E - Why is cluster 17 called aNSCs (3E) when it has the highest GFAP (Fig 3D). Typically, the highest GFAP cells are qNSCs or astrocytes, not aNSCs.

Answer: We previously reported that the level of gfap mRNA expression in neural stem cells (quiescent and activated) did not exactly reflect the amount of protein in these cells. This is the reason why we also used the Slc1a3 marker (Glast), which is highly expressed both at the RNA and protein levels in quiescent NSCs (Daynac et al. 2013).

R2C12: Line 216 - You said in line 216 cluster 13 were astrocytes, then you said in line 227 that cluster 13 was s-qNSC. Which is it?

Answer: This is due to the fact that we performed two distinct analyses.

In the first one (line 216), cells were scored based on datasets provided by Cebrian et al. with one dataset containing genes enriched in astrocytes, and another one, genes enriched in quiescent B-cells. Therefore, cluster 13 was shown to contain 73% cells expressing astrocyte markers, whereas cluster 4 gathered cells expressing both qNSC (B-cells, 48%) and astrocyte (52%) genes.

In the second one (line 227), cells were scored using our transcriptomic signatures of FAC-sorted SVZ cells, which do not include differentiated astrocytes. We demonstrated that the cluster 13 cells only expressed s-qNSC genes.

R2C13: Line 214 - While other clusters were all named in lines 214-221 that were then further discussed in lines 227-230, clusters 15 and 19 were not. You associate both of those clusters with s-iNB - what was it associated with in the above section?

Answer: Lines 219-221 have been reworded as follows: Clusters 10, 5, 15, 12, and 8 were defined as cycling progenitors based on the expression of proliferative markers such as Top2a, Mki67, Ascl1. Clusters 1, 3, 7 and 9 were identified as mNB due to the loss of Mki67, Top2 a and Ascl1 expressions and the expression of Robo2 and Dcx. Cluster 19 that have lost Ascl1 but still expressing Top2a and Mki67 together with Robo2 and Dcx appears at the transition between iNB and mNB.

R2C14: Fig 3I-J - 5 days after irradiation, I would like to see from tissue slices how many cells are dividing compared to 1day post-irradiation and controls. In other paradigms, such as temozolomide experiments (Kalamakis et al), by 5 days we should see less cells in quiescence and more of those quiescent cells exiting quiescence into the cell cycle. Why would there be more cells in quiescence in the irradiated brain? Even if they are radiation resistant, the base number should be comparative between controls and irradiated, which is not what you show in Fig 3I-J.And R2C14Line 234-235 - the text says normalized to numbers of qNSCs which is supposed to be the same (which I agree should be the same). However, your graph in 3I and J shows more qNSCs in irradiated conditions, which would influence greatly and is currently hard to interpret.

Answer: As stated by the reviewer, there is no increase in the absolute number of quiescent cells in the irradiated SVZ. The reconstitution of SVZ cell populations after 4Gy irradiation has already been studied by our group (Daynac et al. 2013, see Fig. 3F), showing that s-iNB and s-mNB are still under-represented after 5 days, while qNSC are in similar numbers as in unirradiated SVZ. Therefore, this led to an over-representation of quiescent cells and early SVZ progenitors in Figure 3J as compared in Figure 3I.

R2C15: Fig 6A - the authors show a significant difference in neurospheres between eGFP- (DCX-) and eGFP+ (DCX+) iNBs - as would be expected as DCX suggests a further commitment towards neurogenic fates, yet your population doubling is the same.

Answer: To determine the population doublings, the medium was changed and cells numbered every 7 days. This condition masked the differences between two cell populations reaching the plateau phase at different time, explaining why eGFP-iNB and eGFP+iNB could not be clearly distinguished by this technique.

R2C16: Fig 6C - Differentiation data (in-vitro) should be quantified in 6C, just as was mentioned for 1C. These values should be done for both of the populations (eGFP-iNB, and eGFP+iNB) and not just compared to the previous pictures which were on total iNB. Again, numbers are required, not just picture examples.

Answer: Quantitative data have been given in Figure 6D showing that approximately 60-80% of cells eGFP+iNB are able to differentiate in either neurons, oligodendrocytes or astrocytes. We did not analyze the differentiation of eGFP-iNB since it would not add any supplementary information.

R2C17: Fig S8 - The authors did not show if the lack of engraftment of eGFP+ cells is due to the transplant (previously you showed only 2/3 worked in a similar paradigm). It would be helpful if the authors would have some means to visualize the DCX low cells to confirm they worked as before in the transplantation (another color? Another type of mouse (Thy1 antigen differences)?) Answer: Unfortunately, the Thy1 antigen has not been documented in mouse subventricular zone progenitors, but only in neurons (PMID: 10813783). Thy1 antigen has also been described in bipotent glial progenitor cell (GCP) from the developing human brain giving rise to oligodendrocytes (PMID: 36931245).

As shown, in Figure S10 we have performed 5 grafts with s-iNB eGFP+ cells, 2 alone and 3 mixed with eGFP- cells and never found any eGFP+ cells 5 weeks after grafting. Moreover, we did not find any eGFP+ cells in the brains of 3 other animals 2 weeks after grafting with s-iNB eGFP+ cells (These data have been added to Figure S10). As compared to the results described in Figure 1 this clearly shows that iNB DCXhigh are not able to generate persistent cells in the grafted brains similarly as mNB.

R2C18: Fig S8 - Why were there no eGFP cells even at the injection site? DCX expression promotes migration, indeed DCX expression becomes very high in cells in the SVZ as they begin to exit to go to the migratory stream. If one didn't see migration, one would expect you would still have survival. Currently, the authors show no cells at 5 weeks, however, they would need to show earlier timepoints as well to determine what is happening with these cells. It is possible these GFP+ cells are not even expressing DCX anymore (see above).

Answer: As stated above, we did not find any GFP+ cells in the brains of 3 other animals 2 weeks after grafting with s-iNB eGFP+ cells (see Figure S10).

R2C19: Line 320 - the authors suggest a subpopulation of NEURONS continues to divide and cite 2 works from the 1990s showing proliferating SVZ cells can differentiate. Our knowledge of this system has come dramatically forward since the 1990s as well as technologically, and to date, neurons have not been shown to divide.

Answer: We apologize for this lack of clarity, as we agree that neurons correspond to differentiated non-cycling cells, but we used the terminology used in these articles. The incorrect part of the sentence Line 320 has thus been deleted from the text.

R2C20: Fig 7 - The whole figure is based on changing levels of RSR genes which were not confirmed in any way to be involved in any of these stages, only descriptively in single-cell analyses.

Answer: As stated above, in our opinion, further characterization of the involvement of RSR genes in neurogenesis is largely beyond the scope of our manuscript. Nevertheless, we think that the role of RSR genes in neurogenesis is an important question that should be addressed in further studies.

Overstatement of findingsR2C21: Fig 1 - Authors did not compare all cell types in each condition but made overstatements about their relationships to each other between graphs. There should also be separate graphs showing all cell types at 4% and a separate one at 20%.

Answer: In the revised version, Figure 1 shows the graph comparing all cell types at 4%O2 and a separate one at 20% as requested by the reviewer. The graphs clearly shows that 4%O2 promotes iNB proliferation compared to the 20% condition.

R2C22: Fig 1D-b2 - Why does DCX look nuclear? One can't say they are only NSCs if they are GFAP as astrocytes also express GFAP. The authors would need another marker to separate those populations. In the text, the authors say expressing GFAP (line 124) which means NSC, but then in line 127 expressing GFAP means astrocytes - which further shows you need additional markers to validate those 2 different cell types.Answer: DCX nuclear translocation has been shown to improve cellular proliferation (PMID:32050972).

As indicated in R1C4. The text has been modified as follows: “Importantly, eGFP+ cells were present in the SVZ of all the animals transplanted with s-iNB eGFP+ and s-NSC/TAP eGFP+ (Fig. 1Db, 1Dc), some of them expressing GFAP indicating the generation of astrocytes, and therefore possibly NSC”.

R2C23: Fig S2 - The transcriptome signature for s-iNBs is very similar to s-TAP, basically suggesting the iNBs are further along in cell cycle.

Answer: This is now the Figure S3. Functional enrichment analysis of individual transcriptome signatures revealed that both s-TAP and s-iNB are enriched in genes related to the cell cycle although with different GO terms enrichments. Indeed, s-TAP are enriched in genes related to G1, G1/S and S phase (but with low -log10 adjusted p-values) and s-iNB with genes related to cell cycle mitosis and M phase (with high -log10 adjusted p-values).

We have previously shown that around 33 % s-iNB have DNA content>2N, versus around 26% of s-TAP and s- aNSC (Daynac et al. 2013), which is in accordance with GO terms enrichments. However, these data have also shown that most s-iNB and s-TAP are in G1, indicating that siNB are not just further along mitosis than TAP.

Moreover, our transcriptomic data clearly show that s-iNB are distinct from s-TAP: (1) according to principal component analyses (Figure 2B et C), the whole transcriptome of s-TAP is closer to that of s-aNSCs than to that of s-iNB (10% variations in PCA2), (2) the heatmap in Figure 2D shows that they have different RSR genes expression profiles, (3) the new Figure S4 shows that GO annotations of differentially spliced isoforms further confirmed that s-TAP and s-iNB have distinct features, and (5) Figure S5 shows that s-iNB expressed genes associated to either TAP or NB that have been described in previous studies, whereas s-TAP did not express genes associated to NB, but look closer to aNSC. Finally, scRNAsq cell clusters related to s-iNB are distinct from the cluster related to s-TAP as shown (1) in Figure 3D and (2) in Figure 4.

R2C24: Fig 3 - The lack of information about timepoint 0 after irradiation, and when proliferation and cell cycle entry begins again following irradiation, limits our interpretation of the single-cell irradiated data.

Answer: We have previously reported the relative abundance of each SVZ neural progenitors in the young adult mouse brain in several papers. Particularly, we based our interpretation on our SVZ irradiation model reported in Daynac et al. 2013 demonstrating a radio resistance of qNSC re-entering into the cell cycle as early as 2 days after 4Gy irradiation successively regenerating aNSC, TAP then iNB and mNB.

R2C25: Fig S3 - These results effectively show that the s-aNSCs and s-TAPs are actually less specific when compared to that same identity in other studies, and that the iNBs are most similar to mitotic TAPs. This supports what was mentioned above, which is that the transcriptional signatures are very similar between the s-TAPs and i-NBs, showing these are not a unique cell state, but just a bit further along mitosis within the TAP cell state.

Answer: This is now the Figure S5. In this figure, we show that s-iNB expressed genes associated to either TAP or NB that have been described in previous studies, whereas s-TAP did not express genes associated to NB, but look like closer to aNSC. As indicated above in R2C23, s-iNB are not just a bit further along mitosis within the TAP cell state. Indeed, we give several data showing that s-iNB and s-TAP have different transcriptomic profiles.

R2C26: Fig 4B - The focus on Ptbp1 as being associated with the iNB cluster border to mNB is expected as all previous studies of Ptbp1 have focused on its role in the progression of other cell types through the cell cycle, its control of cell cycle regulators, and a cell cycle mRNA regulon (Monzon-Casanova et al, 2018, 2019, 2020). This further supports these analyses are specifically defined by cell cycle stages.

Answer: We totally agree that Ptbp1 expression distinguishes cycling cells from postmitotic neuroblasts in accordance with previously published paper, and that based on this unique gene we cannot find any differences between cycling cells ie. aNSC, TAP and iNB. However, as shown in the manuscript and stated above (R2C23 and 25), these cells can be distinguished by their respective expression of many other genes, including other RSR genes.

R2C27: Line 281-282 is an overstatement - the authors suggest that this is a new type of cycling neural progenitor - when all studies point to it being the end of mitosis TAPs as they go on their way to mNBs. This clearly shows a trajectory and not a defined, binary cell type.

Answer: We agree with this statement that the use of the word "type" was misleading, and changed it to "stage" to better reflect that s-iNB are a distinct stage along the differentiation process according to our pseudotime cell-trajectory analysis.

**Author response image 2. sa3fig2:** Pseudotime analysis using Monocle 3 (excluding the cluster 13 corresponding to astrocytes and starting from s-qNSC) revealed two branches starting from s-TAP, one towards cell cycle the other towards neuronal differentiation.

minor comments:R2C28: Fig 3D - For ease, please define what you called the clusters in 3D - not just cluster numbers

Answer: We chose not to call the clusters in 3D because their identification (Group names) is based on data presented after in Figures 3E, F and G.

R2C29: Fig 3E-F - Show astrocytes by text in 3E and F

Answer: As discussed above, astrocytes cannot be shown in these figures because they are based on our signatures which did not include astrocyte signature.